# Cellular and gene signatures of tumor-infiltrating dendritic cells and natural-killer cells predict prognosis of neuroblastoma

Ombretta Melaiu[1], Marco Chierici [2], Valeria Lucarini[1], Giuseppe Jurman [2], Libenzio Adrian Conti [3], Rita De Vito[4], Renata Boldrini[4], Loredana Cifaldi[5], Aurora Castellano[1], Cesare Furlanello [2,6], Vincenzo Barnaba [7,8], Franco Locatelli[1,9] & Doriana Fruci [1✉]

Tumor-infiltrating lymphocytes play an essential role in improving clinical outcome of neuroblastoma (NB) patients, but their relationship with other tumor-infiltrating immune cells in the T cell-inflamed tumors remains poorly investigated. Here we show that dendritic cells (DCs) and natural killer (NK) cells are positively correlated with T-cell infiltration in human NB, both at transcriptional and protein levels, and associate with a favorable prognosis. Multiplex imaging displays DC/NK/T cell conjugates in the tumor microenvironment of low-risk NB. Remarkably, this connection is further strengthened by the identification of gene signatures related to DCs and NK cells able to predict survival of NB patients and strongly correlate with the expression of PD-1 and PD-L1. In summary, our findings unveil a key prognostic role of DCs and NK cells and indicate their related gene signatures as promising tools for the identification of clinical biomarkers to better define risk stratification and survival of NB patients.

[1] Department of Paediatric Haematology/Oncology and of Cell and Gene Therapy, Ospedale Pediatrico Bambino Gesù, IRCCS, 00146 Rome, Italy. [2] Fondazione Bruno Kessler, 38122 Trento, Italy. [3] Confocal microscopy, Core Facility, Research Laboratories, Ospedale Pediatrico Bambino Gesù, IRCCS, 00146 Rome, Italy. [4] Department of Pathology, Bambino Gesù Children's Hospital, 00165 Rome, Italy. [5] Academic Department of Pediatrics (DPUO), Ospedale Pediatrico Bambino Gesù, IRCCS, 00146 Rome, Italy. [6] HK3 Lab, 20129 Milan, Italy. [7] Cellular and Molecular Immunology Unit, Dipartimento di Scienze Cliniche, Internistiche, Anestesiologiche e Cardiovascolari, Sapienza University of Rome, 00161 Rome, Italy. [8] Istituto Pasteur Italia-Fondazione Cenci Bolognetti, 00161 Rome, Italy. [9] Department of Pediatrics, Sapienza University of Rome, 00161 Rome, Italy. ✉email: doriana.fruci@opbg.net

The tumor microenvironment (TME) is a complex network of malignant and non-malignant cells, including stromal cells and immune cells, playing a relevant role in cancer development. Histological analyses on a large collection of human tumors allowed the identification of immune T-cell subsets with either favorable or deleterious effects on clinical outcome[1–3]. The high density of tumor-infiltrating CD8+ T cells has been associated with improved clinical outcomes in many types of solid cancers[4,5].

A robust infiltration of cytotoxic T cells has been associated with the abundance of conventional type 1 dendritic cells (cDC1s), a subset of DC able to stimulate naive tumor antigen-specific CD8+ T cells inside the tumor[6]. This immune cell subtype expresses typical markers of DCs and specific transcriptional signatures enriched of genes involved in the tumor antigen processing and cross-presentation, co-stimulation, and expression of chemokines promoting recruitment of tumor-reactive CD8+ T cells[7]. Within tumors, cDC1s stimulate and expand tumor-specific effector T cells, supporting their reactivation by secreting interleukin-12 (IL-12)[8,9]. Consistently, a high number of cDC1s has been detected in spontaneous regressing tumor models, suggesting that they may be critical for robust tumor control.

The importance of cDC1s in T-cell-mediated antitumor immunity has been established in different experimental mouse tumor models. Indeed, the absence of cDC1s in mice lacking the transcription factor Batf3[10] abolished the ability to reject transplantable immunogenic tumors and to support adoptive T-cell therapy and immune-checkpoint blockade[8,11–14]. Despite the involvement of cDC1s in antitumor immunity[6,8,12,14], their prognostic role has been poorly investigated in humans[8,15–18].

We have previously reported that tumor-infiltrating T cells (TILs) have a prognostic value in neuroblastoma (NB)[19], the most common extracranial solid tumor of childhood, which accounts for 15% of pediatric cancer deaths[20]. NBs associated with good prognosis were characterized by a higher number of proliferating T cells and a more structured T-cell organization, which was gradually lost in highly aggressive tumors[19]. However, the relationship of these cells with other tumor-infiltrating immune cell populations in the T-cell-inflamed NB remains poorly investigated.

Given the importance of cDC1s in cancer immunity, we sought to evaluate whether these or other immune cell populations have a prognostic role in NB patients. Here, we show that T-cell infiltrated NBs are enriched with both DCs and NK cells, and that their abundance is positively correlated with favorable clinical outcome of NB patients. In addition, by exploring our Nanostring and public RNA SEQC expression profiles, we identified two specific gene signatures related to DCs and NK cells able to strongly predict the survival of NB patients. These latter results were also supported by data obtained across multiple adult cancer types.

## Results

**CD3E expression correlates with an enrichment of immune-stimulatory molecules in human neuroblastoma.** To identify potential targets that stimulate T-cell-mediated antitumor immune responses in NB, we assessed the transcriptomes of 498 human primary NB patients from the US-FDA SEQC Project (SEQC-NB) available in the Gene Expression Omnibus (GEO) repository[21]. We found that most of the 757 analyzed immune genes were differently expressed (Supplementary Data 1) both with varying CD3E expression and based on hierarchical clustering of NB patients (Fig. 1a and Supplementary Fig. 1a). Reactome pathway analysis uncovered a vast number of functional categories significantly associated with both innate and adaptive immune responses, including interleukin signaling, Toll-like receptor cascade, and immunoregulatory interactions between lymphoid and non-lymphoid cells (Supplementary Fig. 1b and c, Supplementary Data 2). Similarly, Gene Ontology enrichment analysis performed on transcripts upregulated in tumor samples with high CD3E expression revealed an overrepresentation in genes controlling TCR signaling, T-cell activation, co-stimulation, and cell adhesion molecules signaling (Fig. 1b). Among those genes, CD3E mRNA was strongly associated with gene transcripts that control the trafficking of intratumoral DCs and effector T and NK cells, including FLT3LG, IL2RB, CCL5, IL7R, CXCR3, CXCR6, CCR7, and CCL21[6,15,16,22–25] (Fig. 1c). Consistently, further analyses revealed a positive correlation between CD3E mRNA and individual transcripts related to T-cell activity, including GZMH, GZMK, CTLA4, and TIGIT[6] (Fig. 1d), and those controlling the function of intratumoral DCs (LAMP3 and BTLA)[8,26] and NK cells (KLRK1 and KLRC2)[27] (Fig. 1e). These gene associations were validated in our independent cohort of 36 primary NB samples (Nanostring-NB) by Nanostring gene expression analysis (Supplementary Fig. 2, Supplementary Data 1 and 2).

Altogether, these data indicate that the relative levels of CD3E expression, as an estimate of the abundance of T cells, correlate with an enrichment of immunostimulatory cytokines and chemokines involved in the recruitment of DCs and NK cells in the TME.

**CD3E expression correlates with the levels of intratumoral DC and NK gene signatures in human neuroblastoma.** Using previously validated specific immune gene signatures[28] as an estimate of the immune cell abundance, we found that NB samples with high levels of CD3E (median split) of both Nanostring-NB and SEQC-NB cohorts were significantly enriched of different lymphoid and myeloid cell populations (Fig. 2 and Supplementary Fig. 3, Supplementary Data 3). As expected, immune gene signatures of T-cell subsets including (cytotoxic) CD8+ T cells, (helper) CD4+ T cells, gamma delta T cells, and memory T cells were significantly more abundant in CD3E^high than in CD3E^low NB samples. In addition, the gene signatures of DCs and CD56^dim NK cells were also significantly associated with CD3E expression. By contrast, gene signatures of macrophages and neutrophils were weakly abundant in CD3E^high NB samples, whereas those associated with Tregs, eosinophils, or mast cells displayed discordant results in the two cohorts analyzed (Fig. 2 and Supplementary Fig. 3). Notably, with the exception of macrophages, the gene expression of all immune cell populations enriched in CD3E^high NB samples was abundant in samples expressing CD8A (Supplementary Fig. 4). Differently, NB samples with high CD4 expression were enriched in gamma delta T-cell-, DC-, macrophage-, neutrophil-, and mast cell-related gene expression (Supplementary Fig. 5).

**Intratumoral DCs correlate with increased T-cell infiltration and survival in human neuroblastoma.** Given the strong association between CD3E and the DC gene signature (Fig. 2), we sought to determine the prognostic role of tumor-infiltrating DCs in NB patients. Intratumoral DCs have been previously characterized for the expression of the integrin CD103 in mice and Thrombomodulin (THBD, encoding CD141) in humans[7,11]. A significant correlation was detected between the expression of THBD and the DC gene signature in both Nanostring-NB and SEQC-NB cohorts (Fig. 3a). RNAscope analysis, performed to visualize the cell types expressing THBD, showed that THBD transcript was highly expressed in tumor-infiltrating immune cells resembling myeloid morphology in NB tissues (Fig. 3b). Of note, NB patients with high THBD mRNA correlated with high

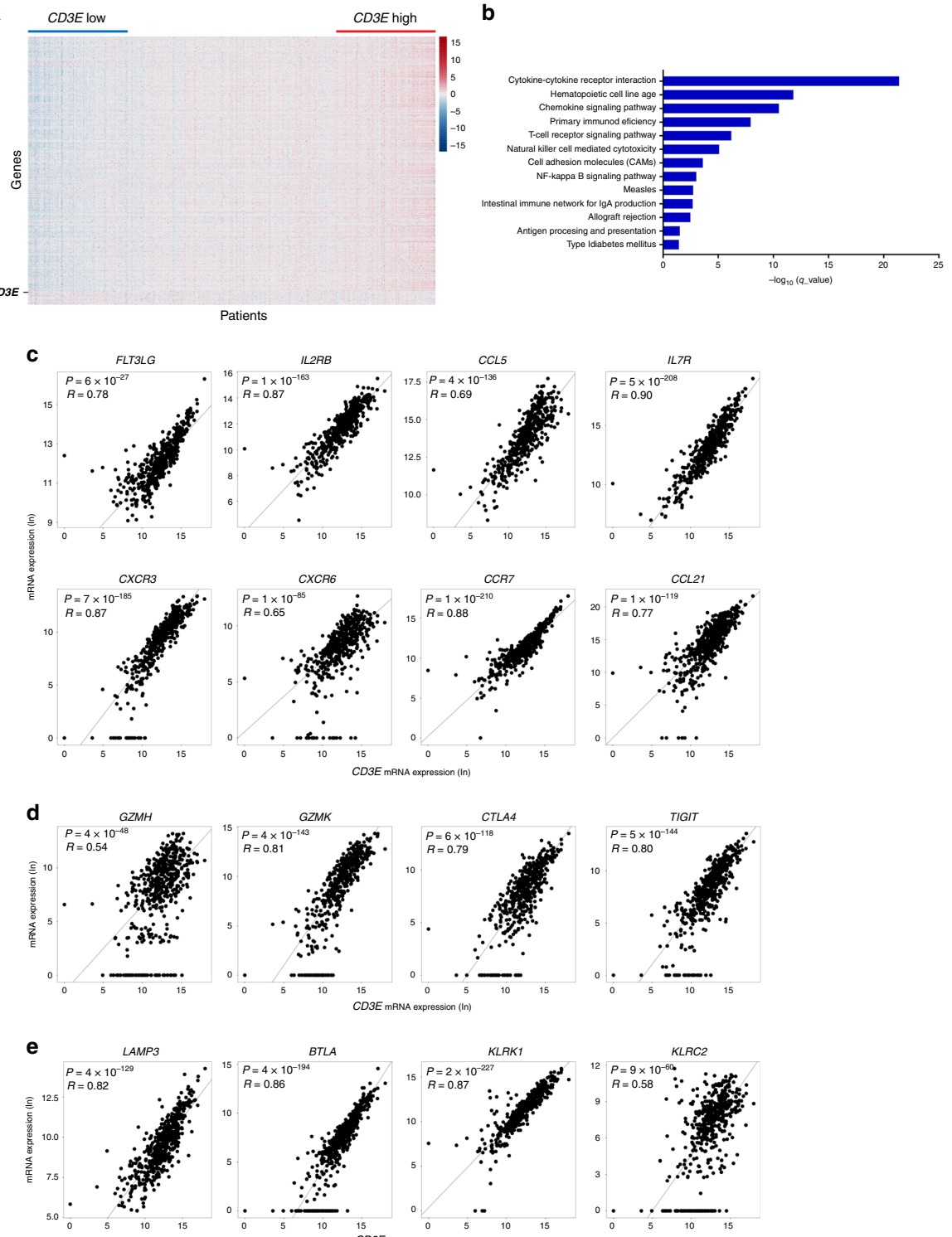

**Fig. 1 *CD3E* expression is associated with antitumor immune responses in human neuroblastoma. a** Heatmap of the normalized expression of immune genes ordered from left to right by increasing levels of *CD3E* expression of 498 NB patients from the SEQC-NB available in the GEO database. **b** Gene ontology term enrichment analysis performed by DAVID Bioinformatics Resources (https://david.ncifcrf.gov/) reveals 13 statistically significant (Benjamini–Hochberg adjusted *P* value <0.05) biological processes controlled by upregulated genes among patients with high *CD3E* expression. **c–e** Correlation of *CD3E* expression with the indicated genes in NB patients. Gene transcripts reported to be associated with increased immune cell infiltration, immune cell trafficking, and immune functional status[15,80–82] were studied in the SEQC-NB dataset. Genes that met the following criteria: (i) a correlation coefficient (R) greater than or equal to at least 0.4 with statistical significance (*P* < 0.05), (ii) a significantly different expression between patients with high and low *CD3E* expression, and (iii) showing consensus in the Nanostring cohort, were displayed. Robust F-test (two-sided) on the robust regression fit of a linear model was used for data analysis. **c** Cytokines and chemokines involved in immune cell trafficking. **d** Activation and exhaustion molecules. **e** DC and NK-cell function markers. Correlations were assessed measuring the coefficient of determination of a robust linear regression model fit on the data (see "Statistical analysis"). No adjustment was required unless otherwise stated. Statistically significant *P* values are indicated.

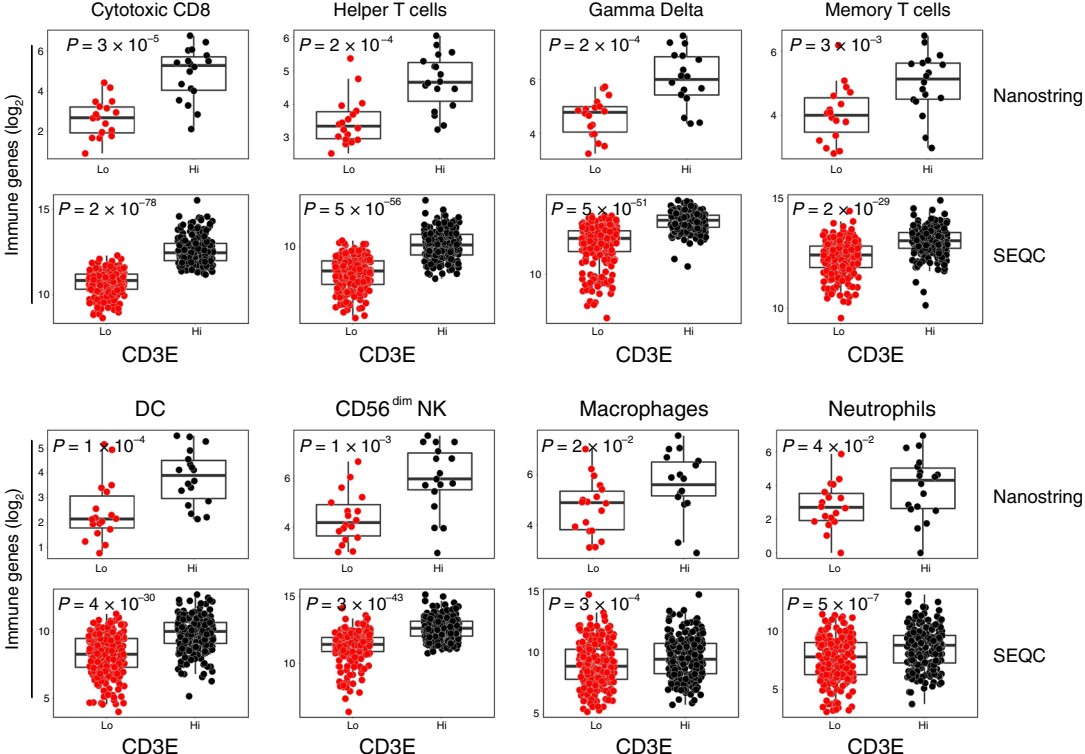

**Fig. 2 Intratumoral DCs and NK cells correlate with CD3E gene expression in human neuroblastoma.** Box plots of metagene expression values for the indicated immune cell types[28] according to the high (Hi) and low (Lo) levels of *CD3E* mRNA (median split) in primary NB patients from Nanostring-NB ($n = 36$) and the SEQC-NB ($n = 498$) cohorts. The immune cell type scores were calculated as the average expression values of the NanoString constituent genes. The boxes show the 25th to 75th percentile; the horizontal lines inside the box represent the median; the upper whisker extends to the largest data point, no more than 1.5 times the IQR from the box; the lower whisker extends to the smallest data point at most 1.5 times the IQR from the box; the dots are individual samples. Data were analyzed by Kruskal–Wallis rank-sum test (two-sided). No adjustment was required. Statistically significant *P* values are indicated.

levels of *CD3E* transcript (Fig. 3c and Supplementary Fig. 6a), and were significantly associated with better survival rates (log-rank *P* values: $9.89 \times 10^{-4}$ and $9.41 \times 10^{-3}$ for overall survival and event-free survival, respectively) (Fig. 3d and Supplementary Fig. 6b). In addition, in combination with the abundance of CD3+ T cells, high levels of both DC and T cells were significantly associated with better prognosis in NB patients (log-rank *P* values: $5.17 \times 10^{-10}$ and $2.99 \times 10^{-5}$ for overall survival and event-free survival, respectively) (Fig. 3e and Supplementary Fig. 6c).

To further examine the clinical significance of *THBD* expression in NB, we performed immunohistochemistry (IHC) analysis on a cohort of 104 NB samples (which included the Nanostring-NB samples), previously characterized for the density of T cells and expression of HLA class I and the immune-checkpoint molecules PD-1, PD-L1, and LAG3[19,29]. CD141 showed an intense membrane staining of cells with myeloid morphology that were either sparsely distributed or localized within the tumor-associated tertiary lymphoid structures (TLS) (Fig. 3f). Fifty-three specimens (51%) had, on average, less than one CD141+ immune cell per mm² (Supplementary Fig. 6d). Among them, 60% (32 samples) belong to the group of high-risk NB patients (with aggressive features including MYCN amplification, presence of metastasis, age at diagnosis >18 months and unfavorable histology) (Supplementary Table 1). Of note, 90% of the deceased patients (16 out of 18) felled within this group (Supplementary Fig. 6d). Consistently with mRNA data, the density of CD141+ cells significantly correlated with the abundance of CD3+, CD8+, and CD4+ T-cell subsets (Fig. 3g and Supplementary Fig. 6e, f), as well as with patient outcome (log-

rank *P* value 0.029) (Fig. 3h). The distribution of CD141+ cells was significantly different in NB patients stratified according to the INRG stage, with higher CD141+ cell density correlating with favorable clinical outcome (Supplementary Fig. 6g). A significant inverse correlation was detected between CD141+ cell density and MYCN amplification status (Wilcoxon rank-sum *P* value: 0.0039), but not with age at diagnosis (Supplementary Fig. 6h, i). Similarly to transcriptomic data (Fig. 3e), high density of CD141+ and CD3+ T cells was significantly associated with improved clinical outcomes of NB patients (log-rank *P* value: 0.0025) (Fig. 3i). In addition, a logistic regression model was built to predict T-cell infiltration from the expression of CD141 by evaluating the predictive performance in terms of Matthews Correlation Coefficient (MCC) over threefold cross-validation (CV) repeated ten times. A significant association was detected between CD141 and CD3 abundance with MCC = 0.50 (95% confidence interval: 0.46–0.55) significantly higher than the random predictor (i.e., a logistic regression model trained after having randomly permuted the target variable; MCC = −0.02; 95% confidence interval: −0.06–0.01) (Wilcoxon rank-sum test $P < 10^{-10}$). Altogether, these results indicate that the relative levels of intratumoral DCs correlate with increased T-cell infiltration and better survival of NB patients.

**Intratumoral NK cells correlate with increased T-cell infiltration and survival in human neuroblastoma.** Then, we sought to determine whether the density of intratumoral NK cells can also predict NB outcome. Expression of the natural cytotoxicity triggering receptor 1 (*NCR1*), a gene encoding the activating receptor

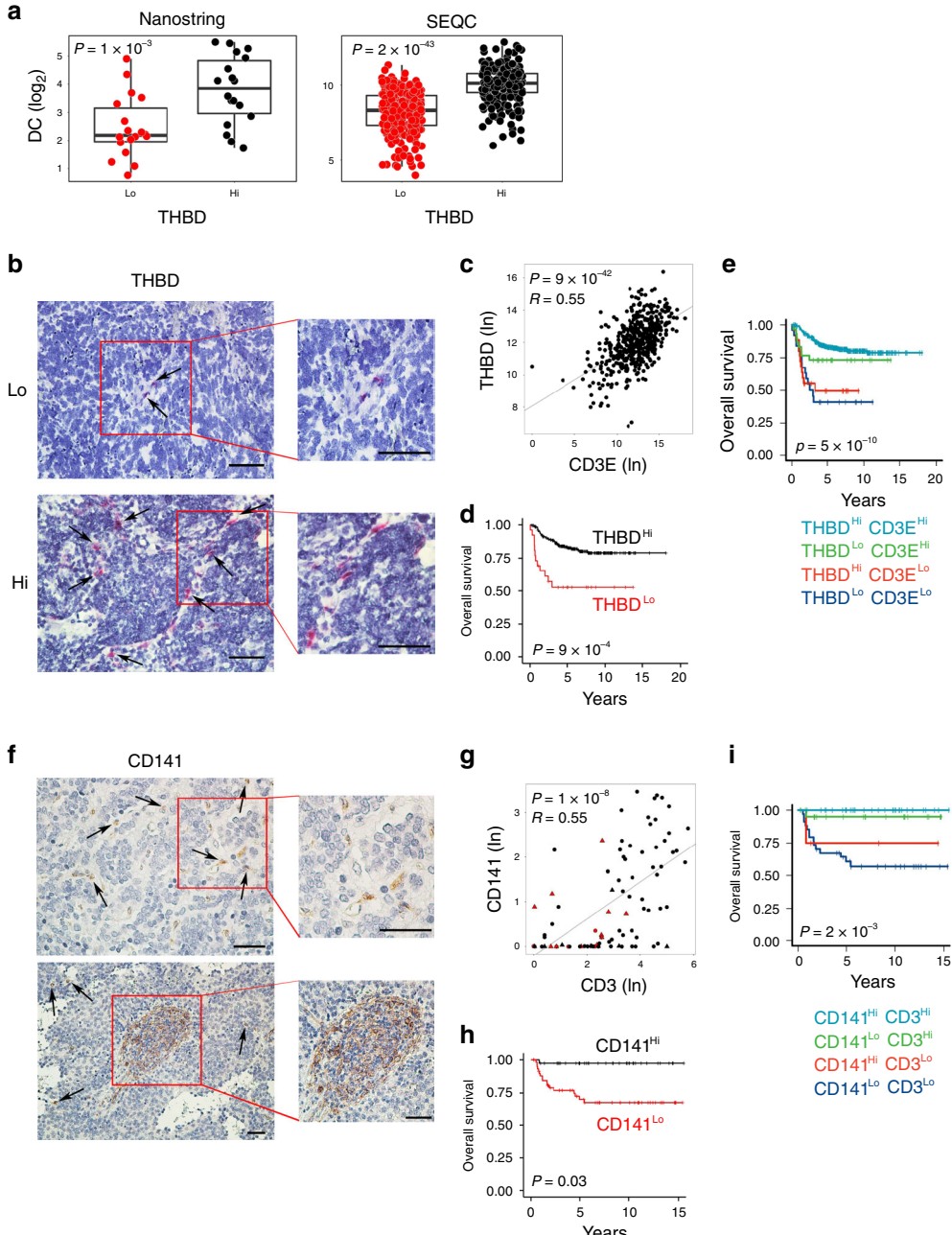

**Fig. 3 Intratumoral DC levels correlate with increased survival in human neuroblastoma. a** Box plots of the DC gene signature[28] according to the high and low levels of *THBD* gene expression (median split) in primary NB lesions from Nanostring-NB ($n = 36$) and SEQC-NB ($n = 498$) cohorts. DC signature score was calculated as the average expression values of the NanoString constituent genes. The boxes show the 25th to 75th percentile; the horizontal lines inside the box represent the median; the upper whisker extends to the largest data point, no more than 1.5 times the IQR from the box; the lower whisker extends to the smallest data point at most 1.5 times the IQR from the box; the dots are individual samples. Data were analyzed by Kruskal–Wallis rank-sum test (two-sided). **b** Representative images of NB tissue sections analyzed for *THBD* expression using RNAscope (red dots). Nuclei were counterstained with hematoxylin (blue). *THBD* positive cells are indicated by black arrows. Original magnification ×40 (left) and ×60 (right). Scale bar, 30 μm. **c** Correlation between *THBD* and *CD3E* gene expression in NB patients, analyzed by robust F-test (two-sided) in SEQC-NB ($n = 498$). **d**, **e** Kaplan–Meier curves show the duration of overall survival of NB patients according to the *THBD* gene expression either alone (**d**), or in combination with *CD3E* gene (**e**) in SEQC-NB ($n = 498$). Log-rank test with Miller and Siegmund *P*-value correction was used. **f** Representative examples of CD141 staining in primary NB lesions. Upper and lower panels, CD141-expressing cells sparsely distributed within tumor nests or localized within TLS, respectively. Brown, CD141-expressing cells. Nuclei were counterstained with hematoxylin (blue). CD141 positive cells are indicated by black arrows. Original magnifications, ×40 (up left), ×60 (up right), ×20 (down left), and ×40 (down right). Scale bars, 30 μm. **g** Scatter plot showing the correlation between CD141+ and CD3+ cell densities in NB patients, analyzed by robust F-test (two-sided). Black and red dots, patients who are alive and dead, respectively; triangle and dots, MYCN amplified and non-amplified patients, respectively. **h**, **i** Kaplan–Meier curves of overall survival of NB patients ($n = 104$) according to the density of CD141+ cells alone (**h**) or in combination with CD3+ cell densities (**i**). Log-rank test with Miller and Siegmund *P*-value correction was used. Correlations in **c** and **g** were assessed measuring the coefficient of determination of a robust linear regression model fit on the data (see "Statistical analysis"). The representative images in **b** and **f** were selected from $n = 104$ biologically independent NB specimens. Hi high, Lo low. No adjustment was required unless otherwise stated. Statistically significant *P* values are indicated.

NKp46 specific for NK cells (median split), showed a significant correlation with the NK-CD56$^{dim}$ signature (Fig. 4a). RNAscope analysis showed a high expression of the *NCR1* transcript in the immune cells infiltrating NB samples (Fig. 4b). NB patients expressing high *NCR1* mRNA displayed high levels of *CD3E* transcript (Fig. 4c), and better survival (log-rank *P* values: 3.33 × $10^{-14}$ and 4.29 × $10^{-9}$ for overall survival and event-free survival, respectively) (Fig. 4d and Supplementary Fig. 7a). As for *THBD*, high levels of NK and T cells were significantly associated with better clinical outcome of NB patients (log-rank *P* values: 1.06 × $10^{-22}$ and 1.71 × $10^{-10}$ for overall survival and event-free survival, respectively) (Fig. 4e and Supplementary Fig. 7b). Unlike *THBD*, high levels of *NCR1* were associated with better prognosis in patients with either high or low CD3$^+$ T-cell abundance (*NCR1*$^{high}$*CD3E*$^{high}$ vs *NCR1*$^{low}$*CD3E*$^{high}$ pairwise log-rank *P* value = 2.5 × $10^{-8}$ and *NCR1*$^{high}$*CD3E*$^{low}$ vs *NCR1*$^{low}$*CD3E*$^{low}$ pairwise log-rank *P* value = 0.017) (Fig. 4e). These data were confirmed at the protein levels (Fig. 4f–i and Supplementary Fig. 7c–h). IHC analysis on the cohort of 104 NB samples revealed an intense membrane staining of NKp46 in NK cells sparsely distributed within the tumor nests (Fig. 4f). Seventy-two specimens (69%) had less than one NKp46$^+$ cell per mm$^2$. Among them, 51% (37 samples) belong to high-risk NB patients (Supplementary Table 1). Similar to DCs, the majority of the deceased patients (16 out of 18) felled in this group (Supplementary Fig. 7c). The density of NKp46$^+$ cells was significantly correlated with the abundance of CD3$^+$ and CD4$^+$ T-cell subsets, and to a lesser extent with CD8$^+$ T cells (Fig. 4g and Supplementary Fig. 7d, e), and associated with better overall survival (log-rank *P* value: 0.018) (Fig. 4h). The distribution of NKp46$^+$ cells was statistically different in NB patients stratified according to the INRG stage, with a higher density of NKp46 correlating with a favorable prognosis (Supplementary Fig. 7f). Similarly to CD141, the density of NKp46$^+$ cells was significantly inversely correlated with MYCN amplification status (Wilcoxon rank-sum *P* value: 0.0022) (Supplementary Fig. 7g), but not with age at diagnosis (Supplementary Fig. 7h). Similar to mRNA data (Fig. 4e), patients with a high density of NKp46$^+$ and CD3$^+$ T cells were associated with a better prognosis (log-rank *P* value: 0.0026) (Fig. 4i).

Next, a logistic regression model was built to predict T-cell infiltration from the expression of NKp46 by evaluating its predictive performance in terms of MCC over a threefold CV repeated ten times. As for CD141, a significant association was detected between NKp46 and CD3 abundance (MCC = 0.42; 95% confidence interval: 0.39–0.46), significantly higher than the random predictor (MCC = −0.06; 95% confidence interval: −0.13–0.00) (Wilcoxon rank-sum test *P* < $10^{-10}$).

Taken together, these results indicate that the density of intratumoral NK cells, like intratumoral DCs, correlates with increased T-cell infiltration and better survival of NB patients.

**Correlation between intratumoral DC and NK cells in human neuroblastoma.** Given the role of intratumoral DCs and NK cells in predicting the survival of NB patients, we sought to consider whether these two immune cell populations interact and correlate with each other. First, we performed a multiplex immunofluorescence (IF) analysis in T-cell-enriched NB samples to evaluate the spatial distribution of DCs and NK cells within the TME. In line with the in situ IHC analyses[19,29] (Figs. 3f and 4f), we found that these immune cells were detected both within tumor nest and tumor septa regions of T-cell-inflamed NB samples, with DCs directly interacting with NK cells, and/or CD8$^+$ T cells (Fig. 5a and Supplementary Figs. 8–10). Cell counting revealed that CD8$^+$ T cells were the most abundant, followed by DC and

NK cells. Interestingly, 10% of the total CD8$^+$ T cells were found attached to DCs, while 1.2% was in contact with NK cells (Fig. 5a), thus suggesting the existence of spatial interaction between CD8$^+$ T cells, DCs and NK cells in the NB microenvironment. Tumor samples with high levels of *THBD* mRNA (median split) were significantly enriched of CD56$^{dim}$ NK cells in both Nanostring-NB and SEQC-NB cohorts (Fig. 5b), as well as high levels of *NCR1* mRNA were associated with an abundance of intratumoral DCs (Fig. 5c). Consistently, the expression of *NCR1*/NKp46 was significantly correlated with that of *THBD*/CD141 at both RNA (Fig. 5d) and protein levels (Fig. 5e). High expression of *NCR1*/NKp46 was associated with better survival in NB patients regardless of *THBD*/CD141 expression (log-rank *P* values: 2.8 × $10^{-13}$ and 0.026, respectively, for overall survival, and log-rank *P* value: 1.68 × $10^{-8}$ for event-free survival) (Fig. 5f, 5g and Supplementary Fig. 10b). We next investigated the association of *THBD*/CD141 and *NCR1*/NKp46 with MYCN amplification status, age at diagnosis and INRG stage. We built multivariate logistic regression models to predict MYCN amplification, age at diagnosis or INRG from the expression of *THBD*/CD141 and *NCR1*/NKp46 (one at a time, as well as in combination), evaluating their predictive performance in terms of MCC over a threefold CV repeated ten times. For MYCN amplification and age at diagnosis, the logistic regression predictive performance was not significantly different from that of a random predictor (MCC ~0), i.e., a logistic regression model fit on the data after having randomly scrambled the response variable (Wilcoxon rank-sum test *P* = 0.49). A significant association with INRG was found for *THBD*/CD141 and/or *NCR1*/NKp46, with MCC = 0.53 (95% confidence interval: 0.48–0.57), significantly higher than the random predictor MCC = 0.01 (95% confidence interval: −0.04–0.06) (Wilcoxon rank-sum test *P* < $10^{-3}$) (Supplementary Fig. 10c).

Altogether, these results indicate that DCs and NK cells interact and correlate each other in the TME of NB, and that they are significant predictors of clinical outcome independent from the criteria currently used to stage NB (i.e., MYCN amplification status and age at diagnosis). Thus, DCs and NK cells could be usefully integrated as prognostic markers for a better NB risk stratification.

**DC and NK gene signatures predict clinical outcome in human neuroblastoma.** Next, we sought to generate gene signatures that are predictive of the tumor infiltration by DCs and NK cells in NB patients. To this end, we focused on genes that were (i) significantly associated with high levels of *THBD* or *NCR1* (fold change > 2 or < −2, and Q or *P* value < 0.05) from Nanostring data, (ii) with highly significant prognostic value (HR < 1 or >1), and (iii) predominantly expressed by DCs and/or NK cells. Among the 757 immune genes, we found 189 genes upregulated and 9 genes downregulated in association with high levels of *THBD*, and 14 genes upregulated in association with high levels of *NCR1* (Fig. 6a). A Cox proportional-hazard survival analysis identified 146 genes (134 and 12 genes upregulated in association with high levels of *THBD* and *NCR1*, respectively) predicting the good prognosis (HR < 1, FDR < 0.05), and 6 genes (downregulated in association with high levels of *THBD*) predicting the poor prognosis (HR > 1, FDR < 0.05) (Fig. 6a and Supplementary Data 4). The Primary Cell Atlas database from BioGPS[30] revealed that 16 of the 146 genes associated with good prognosis (9 for *THBD* and 7 for *NCR1*) were expressed by DCs and/or NK cells (Fig. 6a, b). Notably, some genes associated with high levels of *THBD* were also (or exclusively) expressed by NK cells (*CXCR1*, *GZMK*, *IL-18R1*, *PRF1*, *SIGIRR*), and vice versa, one of the genes associated with high levels of *NCR1* was exclusively expressed by DCs (*BTLA*) (Fig. 6b and Supplementary Figs. 11 and 12). Gene

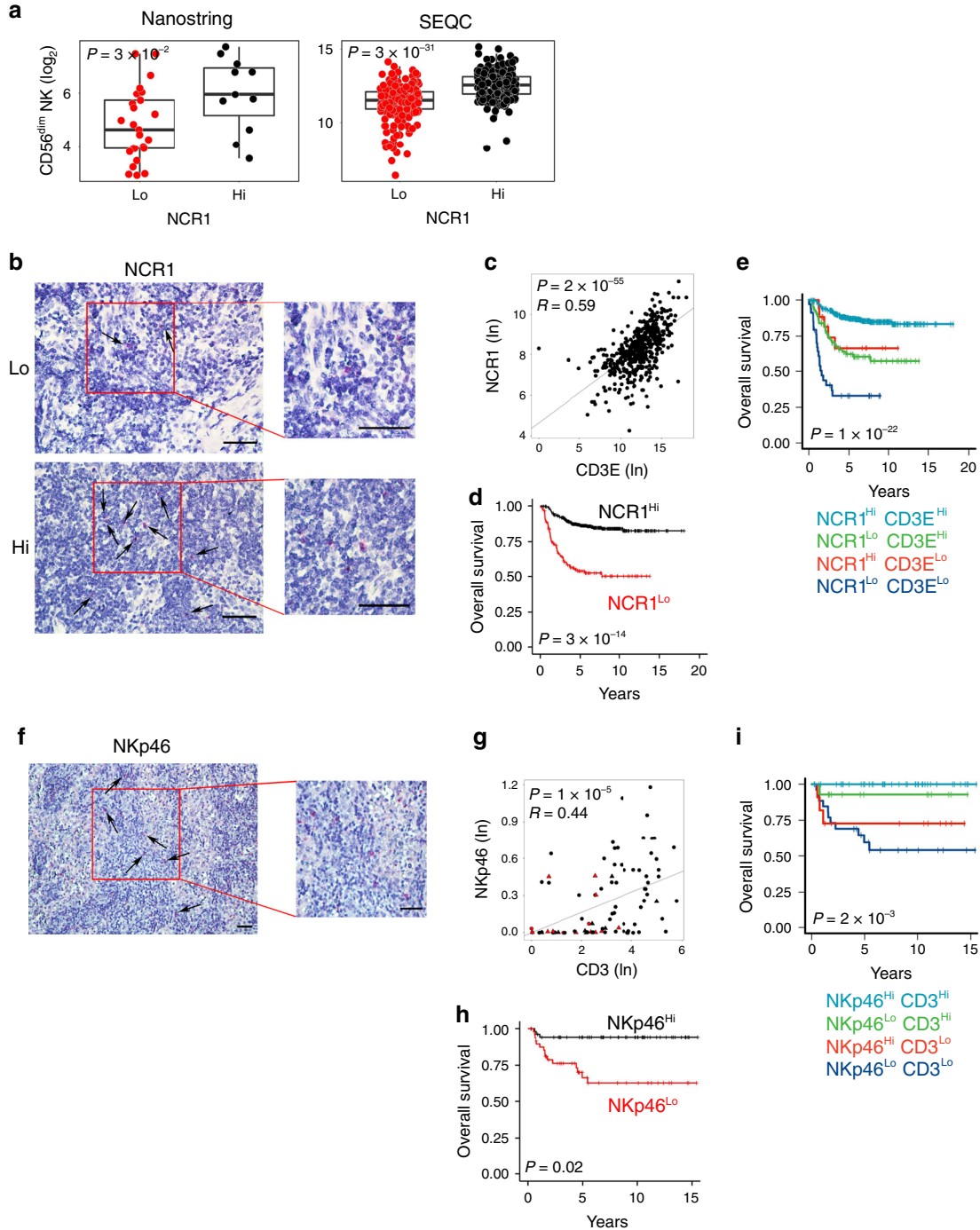

ontology enrichment analysis showed that these genes were mainly involved in T-cell-mediated cytotoxicity (FDR: $2.89 \times 10^{-6}$), NK-cell activation (FDR: $8.4 \times 10^{-5}$), cellular defense response (FDR: $1.26 \times 10^{-4}$) and positive regulation of cytokines involved in immune responses (FDR: $5.77 \times 10^{-3}$), thus suggesting their bona fide functional relevance to the contribution of DCs and NK cells in the antitumor immune responses. Consistently, protein–protein interaction network (STRING) analysis (http://string-db.org) confirmed their functional interplay (average local clustering coefficient: 0.763; PPI enrichment value $1 \times 10^{-16}$; Fig. 6c). Interestingly, none of the six genes predicting poor prognosis was expressed by DCs and/or NK cells (Fig. 6b and Supplementary Fig. 11). This was confirmed by RNAscope analysis performed on three out of six genes (BIRC5, CDK1, and PBK), showing their predominant

expression on tumor cells of NB patients (Fig. 6d). Gene ontology enrichment analysis revealed that BIRC5, CDK1, PBK, and TTK were involved in the cell cycle regulation (FDR: $1.74 \times 10^{-5}$). Interestingly, PBK, TTK, BAGE, and CT45A encode proteins termed cancer/testis antigens known to confer a selective advantage to tumor cells by promoting oncogenic processes or permitting evasion of tumor-suppressive mechanisms[31].

Unsupervised clustering analysis on THBD- and NCR1-associated genes allowed the identification of patients with distinct gene expression profiles and clinical outcome (Fig. 6e). We evaluated the patient survival status after combining the expression of THBD- and NCR1-associated genes and found that high expression of both gene clusters was associated with increased survival in the SEQC-NB dataset, employed as the

**Fig. 4 Intratumoral NK-cell levels correlate with increased survival in human neuroblastoma. a** Box plots of the NK gene signature[28] according to the high and low levels of *NCR1* gene expression (median split) in primary NB lesions from Nanostring-NB ($n = 36$) and SEQC-NB ($n = 498$) cohorts. CD56[dim] NK-cell signature score was calculated as the average expression values of the NanoString constituent genes. The boxes show the 25th to 75th percentile; the horizontal lines inside the box represent the median; the upper whisker extends to the largest data point, no more than 1.5 times the IQR from the box; the lower whisker extends to the smallest data point at most 1.5 times the IQR from the box; the dots are individual samples. Data were analyzed by Kruskal–Wallis rank-sum test (two-sided). **b** Representative images of NB tissue sections probed for *NCR1* expression using RNAscope (Red dots). Nuclei were counterstained with hematoxylin (blue). *NCR1*-positive cells are indicated by black arrows. Magnification with ×40 (left) and ×60 (right). Scale bar, 30 μm. **c** Correlation between *NCR1* and *CD3E* gene expression in NB patients, analyzed by robust F-test (two-sided) in SEQC-NB ($n = 498$). **d**, **e** Kaplan–Meier curves of overall survival of NB patients according to the *NCR1* gene expression either alone (**d**) or in combination with *CD3E* gene (**e**) in SEQC-NB ($n = 498$). Log-rank test with Miller and Siegmund *P* value correction was used. **f** Representative example of NKp46 cell staining in primary NB lesion. Red, NKp46-expressing cells. Nuclei are counterstained with hematoxylin (blue). NKp46-positive cells are indicated by black arrows. Magnification with ×20 (left) and ×40 (right). Scale bar, 30 μm. **g** Scatter plot showing the correlation between NKp46[+] and CD3[+] cell densities in NB patients, analyzed by robust F-test (two-sided). Black and red dots, patients who are alive and dead, respectively; triangle and dots, MYCN amplified and non-amplified NB patients, respectively. **h**, **i** Kaplan–Meier curves of overall survival of NB patients according to the NKp46 protein expression either alone (**h**) or in combination with CD3[+] cell densities (**i**). Log-rank test with Miller and Siegmund *p*-value correction was used. Correlations in **c** and **g** were assessed measuring the coefficient of determination of a robust linear regression model fit on the data (see "Statistical analysis"). The representative images in **b** and **f** were selected from $n = 104$ biologically independent NB specimens. Hi high. Lo low. No adjustment was required unless otherwise stated. Statistically significant *P* values are indicated.

training set for the signature discovery (log-rank *P* values: $3.24 \times 10^{-10}$ and $1.55 \times 10^{-11}$ for overall survival of *THBD*- and *NCR1*-associated genes, respectively, and log-rank *P* values: $4.62 \times 10^{-6}$ and $5.57 \times 10^{-5}$ for event-free survival of *THBD*- and *NCR1*-associated genes, respectively) (Fig. 6f and Supplementary Fig. 13a). In addition, fivefold CV revealed that high expression of both DC and NK gene signatures continues to be significantly associated with better OS (Supplementary Fig. 13b). Of note, when ten iterations of fivefold CV ($10 \times 5$-fold) were further employed to enhance the robustness of the results, OS remains statistically significant for DC and NK gene signatures in 82 and 78% of cases, respectively, and showed the same trend in the remaining cases (Supplementary Data 5)[21].

Next, we assessed the prognostic improvement of *THBD*- and *NCR1*-associated genes with respect to classical predictors such as INRG stage, MYCN amplification status, age at diagnosis, and T-cell infiltration. To this aim, we evaluated the MCC, sensitivity (sens), and specificity (spec) of three logistic regression models predicting the binary outcome OS > 2/OS < 2 yrs on the SEQC-NB cohort, using the classical predictors alone (MCC = 0.503, sens = 0.692, spec = 0.790) as well as in combination with the DC signature (MCC = 0.525, sens = 0.717, spec = 0.799) and the NK signature (MCC = 0.552, sens = 0.741, spec = 0.806). The performance metrics improve with the addition of the DC and NK signatures, thus suggesting their prognostic improvement with respect to classical predictors (Supplementary Fig. 13c and Supplementary Data 6). The extent of the prognostic value of DC and NK signatures was verified on both a further NB dataset (SEQC-NB2)[32] (Supplementary Fig. 14a) and across multiple adult cancer models, including colorectal cancer[33], skin cutaneous melanoma (SKCM), head-neck squamous cell carcinoma (HNSC), and breast cancer[34] (Supplementary Fig. 14b–e).

These findings provide evidence that genes associated with high levels of *THBD* and *NCR1* expression represent robust immune gene signatures predicting both infiltration of DCs and NK cells in tumors and survival of cancer patients.

Notably, the expression of *THBD*- and *NCR1*-associated genes was significantly correlated with genes encoding immune checkpoints PD-L1 and PD-1 in NB samples (Fig. 6g and Supplementary Figs. 15 and 16). Specifically, *CD274* and *PDCD1* genes showed a direct correlation with all the individual genes belonging to the two immune signatures having a protective role (HR < 1), and an inverse correlation with those associated with poor prognosis (HR > 1) (Supplementary Figs. 15 and 16).

Finally, to dissect the relative contribution of the two gene signatures as compared to the other variables known to

associate with patient survival in univariate analysis (namely, INRG L1, L2, and MS, absence of MYCN amplification, age <18 months, high T-cell infiltration and high expression of *THBD* and *NCR1* gene signatures), a multivariate Cox regression analysis was performed. High expression of *THBD* and *NCR1* gene signatures remained statistically associated with better survival (HR, 0.4; 95% confidence interval, 0.27–0.61; $P = 1.4 \times 10^{-5}$, and HR, 0.36; 95% confidence interval, 0.23–0.55; $P = 4.0 \times 10^{-6}$ for *THBD* and *NCR1* signatures, respectively), similarly to the T-cell abundance (HR, 0.35; 95% confidence interval, 0.22–0.55; $P = 5.4 \times 10^{-6}$).

These results indicate that both *THBD* and *NCR1* gene signatures represent novel prognostic variables able to predict overall survival in NB patients.

## Discussion

We previously reported that T-cell infiltration was positively associated with a better prognosis of NB[19,29]. Since T cells are not autonomous in their effector functions, depending on interaction with other cells for the onset and maintenance of T-cell responses, herein, we investigated which other immune cell populations could correlate with the survival of NB patients. We show that tumors highly infiltrated by T cells are also enriched of intratumoral DCs and NK cells that, similarly to T cells, are associated with a favorable clinical outcome of NB patients. Although less abundant than T cells, both DCs and NK cells are spatially distributed within the nest tumor regions interacting with each other, and with CD8[+] T cells. In addition, we found that DCs and NK cells are predictors independent from those currently used to stage and stratify treatment of patients with NB (i.e., MYCN amplification status and age at diagnosis). Notably, by exploring both our Nanostring and public RNA SEQC expression profiles, we identified gene signatures related to DCs and NK cells, which are strongly associated with the survival of NB patients. Intratumoral DCs are particularly important for cancer immunity for their ability in taking up antigens inside the tumor bed, trafficking to the nearest tumor-draining lymph nodes, and performing the cross-priming to activate cytotoxic T-cell responses and eliminate the tumor[6]. Indeed, no tumor-specific CD8[+] T cells can be generated in murine models lacking intratumoral DCs. Experimental cancer models revealed that accumulation and activation of tumor-infiltrating effector CD8[+] T cells are dependent on the presence of intratumoral CD103[+] DCs[8], suggesting that the limited infiltration of CD8[+] T cells

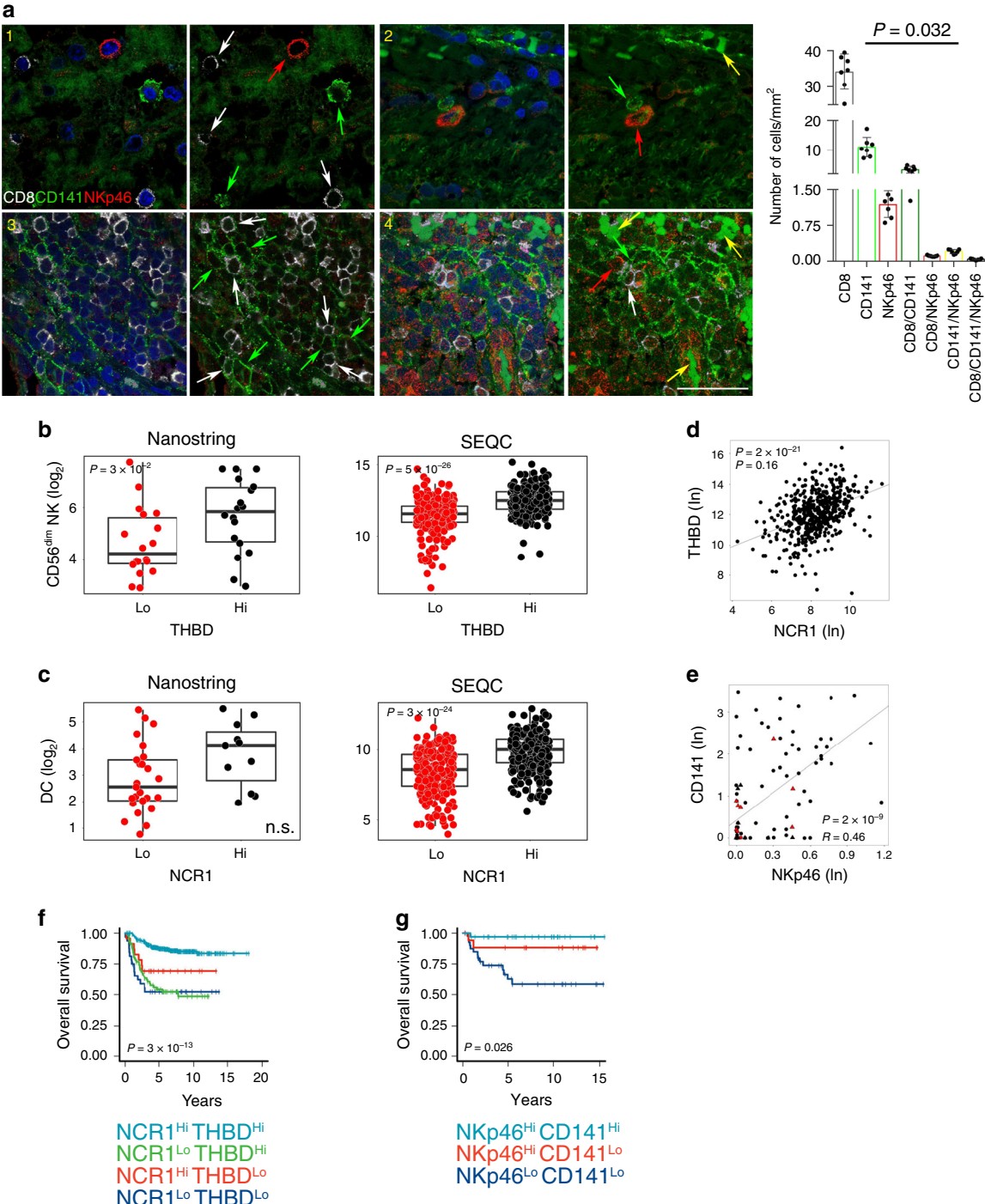

observed in many tumor lesions could be a consequence of the reduced accumulation of this DC subset. Our findings in the context of NB support this concept, showing a direct correlation between the abundance of TILs and DCs both at mRNA and protein levels.

In addition to T cells and DCs, also intratumoral NK cells have a prognostic role in NB patients. NK cells are well known to contribute to antitumor immunity and predict good prognosis in several human cancer patients[35]. Interestingly, we found that tumors with low NK-cell density were also devoid of intratumoral DCs and associated with poor prognosis. Furthermore, the abundance of intratumoral DCs was associated with better survival mainly in patients whose samples were enriched in NK cells.

Of note, recent studies have highlighted the existence of the NK-DC axis in adult tumors, including melanoma, breast cancer, lung cancer, and head and neck squamous cell carcinoma[15,16], and that these innate immune cells correlate with responsiveness to anti-PD-1 immunotherapy[15]. In this context, we found that *FLT3LG* and *CCL5* genes, encoding the related proteins contributing to the recruitment of intratumoral DCs by NK cells[15,16], are both strongly correlated with *CD3E* (Fig. 1), *THBD* and *NCR1* (Supplementary Fig. 17a), thus further validating the existence of the DC–NK axis in NB. Consistently, a direct interaction between the number of TILs, DCs, and NK cells was detected in NB. Indeed, multiplexed immunofluorescence imaging clearly showed that DCs and NK cells interact with each other within tumor nest

**Fig. 5 Cross-correlation between intratumoral DCs and NK cells in human neuroblastoma. a** Multiple immunofluorescence staining of NB tumor lesions for CD8 (white), CD141 (green), and NKp46 (red), shown at magnification ×60 (zoom), scale bar 30 μm. Four representative scenarios of these cells within the tumor are shown: (1) CD8+, CD141+ and NKp46+ cells in close proximity to each other; (2) CD141+ cells interacting with NKp46+ cells; (3) CD141+ cells interacting with CD8+ T cells; (4) CD141+ cells interacting with both CD8+ and NKp46+ cells. Images with nuclei (Hoechst) are shown on the right of each panel. Red blood cells are indicated by yellow arrows. Quantitative analysis of the indicated immune cells from $n = 7$ biologically independent highly infiltrated NBs is shown on the right. Plotted as mean ± S.D. and analyzed by Kruskal–Wallis test to generate two-tailed $P$ values. **b**, **c** Box plots of the NK gene signature according to the high and low levels of *THBD* gene expression (median split) (**b**), and of the DC gene signature according to the high and low levels of *NCR1* gene expression (median split) (**c**) in primary NB lesions from Nanostring-NB ($n = 36$) and SEQC-NB ($n = 498$) cohorts. DC and CD56$^{dim}$ NK-cell signature scores were calculated as the average expression values of the NanoString constituent genes. Kruskal–Wallis rank-sum test (two-sided) was used. The boxes show the 25th to 75th percentile; the horizontal lines inside the box represent the median; the upper whisker extends to the largest data point, no more than 1.5 times the IQR from the box; the lower whisker extends to the smallest data point at most 1.5 times the IQR from the box; the dots are individual samples. **d**, **e** Correlation between *NCR1* and *THBD* gene expression on SEQC-NB dataset ($n = 498$) (**d**) and NKp46+ and CD141+ cell densities ($n = 104$ independent biological specimens) (**e**) in NB patients. In **e**, black and red dots, patients who are alive and dead, respectively; triangle and dots, MYCN amplified and non-amplified patients, respectively. Robust F-test (two-sided) was used for data analysis. **f**, **g** Kaplan–Meier curves of overall survival of NB patients according to the combined expression of *THBD* and *NCR1* gene (**f**) and CD141 and NKp46 protein (**g**) levels on 498 and 104 NB independent biological specimens, respectively. In **g**, due to its low frequency, the group NKp46$^{Low}$-CD141$^{High}$ was omitted. Log-rank test with Miller and Siegmund $P$-value correction was used. Correlations in **d** and **e** were assessed measuring the coefficient of determination of a robust linear regression model fit on the data (see "Statistical analysis"). Hi high, Lo low. No adjustment was required unless otherwise stated. Statistically significant $P$ values are indicated. n.s = not significant.

regions and in proximity to the TLS. Interestingly, previous studies reported that DC–NK crosstalk depends on both cell-to-cell contact and soluble factors[36,37]. Activated NK cells strongly enhance DCs maturation, and improve their ability to stimulate allogeneic naive CD4+ T cells in a cell contact-dependent manner, together with a relevant secretion of IFN-γ and TNF-α[38,39]. Accordingly, several studies have also shown that the presence of mature DCs and T cells within tumor-associated TLS is necessary to orchestrate cytotoxic and immune contexture and that they are associated with good clinical outcome[28,40–44] and response to therapeutic vaccines[45,46].

The existence of the DC–NK axis in NB was further strengthened by the identification of the 16 gene DC (*THBD, AXL, CX3CR1, FOS, GZMK, IL18R1, MEF2C, PRF1, SIGIRR, SYK, BAGE, BIRC5, CDK1, CT45A, PBK,* and *TTK*) and 8 gene NK (*NCR1, BTLA, CD7, CD160, CD247, GZMM, KLRC1,* and *KLRC2*) cell signatures. Indeed, we found that the DC gene signature is enriched of transcripts typically expressed by NK cells (*CX3CR1, GZMK, IL18R1, PRF1,* and *SIGIRR*), and that vice versa the NK gene signature includes the *BTLA* transcript expressed on DCs[47]. Importantly, the detection of the single-gene transcript belonging to the two signatures, taken individually, was sufficient to discriminate between patients with a better or worse prognosis. The prognostic value further increased when genes were clustered all together.

In addition to the genes already known to be associated with T-cell infiltration (*CD7, CD247,* and *SYK*), and DC and NK-cell functions (*BTLA* for DC[8], and *CD160, PRF1, GZMK, GZMM, KLRC1,* and *KLRC2* for NK cells[48]), the two signatures include genes predicting a prognostic benefit in different types of cancer. In particular, high expression of *SIGIRR* has been reported to have a protective role in malignancies that thrive on inflammation[49], including human diffuse large B-cell lymphoma[50], and chronic lymphocytic leukemia[51]. Similarly, an increase of IL-18R1 reduced tumor burden in liver cancer mouse models[52]. Moreover, the use of IL-18 has been proposed as a novel approach to increase DC recruitment to the inflammation site of IL-18R1 expressing tumors[53]. This evidence supports our findings, providing a proof of principle that both DC and NK gene signatures may serve as clinical biomarkers in NB at the same extent of prognostic tools routinely used to stratify NB patients (including T-cell infiltration).

Although infiltration of human cancers by T cells is generally interpreted as a sign of good prognosis, the number of intratumoral CD8+ T cells able to recognize tumor antigens is very limited and variable[54]. Thus, to identify patients potentially benefiting from immunotherapy protocols, it should be considered a more comprehensive scenario by evaluating not only the total T-cell content but also the abundance of DCs and NK cells. In this regard, increasing the recruitment and activation of DCs and NK cells within the TME may represent a successful strategy to restore effector T-cell accumulation and improve the effectiveness of checkpoint blockades and other immunotherapies in tumors not infiltrated by T cells. Consistently, we found a significant overrepresentation of genes encoding for activation and cytotoxicity molecules, such as *IFNG, PRF1, GZMB, GZMH, GZMK, GZMA, IL7R,* and *ICOS,* among patients with high levels of *CD3E, THBD,* and *NCR1,* as compared to those with low expression of these genes (Supplementary Fig. 17b, c).

Besides T-cell infiltrate and neoantigen load[55,56], other factors have recently been proposed as biomarkers to predict the effectiveness of immunotherapy, including PD-L1 expression[57], antigen presentation defects[58], interferon signaling[59], mismatch repair deficiency[60], tumor aneuploidy[61], and gut microbiota[62]. However, none of these factors is sufficient individually to achieve accurate outcome prediction[57]. In this context, the identification of specific transcriptome signatures to be employed as reliable immunotherapeutic biomarkers may represent a major challenge[63]. To date, specific gene-panels, such as Oncotype DX[64], MammaPrint[65], and Prosigna[66], have demonstrated clinical utility in predicting treatment benefits in breast cancer. Interestingly, in NB samples, the targets of current checkpoint blockade therapies PD-1 and PD-L1 showed a significant correlation with the DC and NK gene signatures. Further studies on the prospective data of immune-checkpoint inhibitor-treated patients are needed to determine whether such gene signatures could be exploited as a biomarker predicting patients' survival or response to immunotherapies, in the clinical practice.

Overall, our findings highlighted that the presence of DCs and NK cells is positively correlated with both T-cell infiltration and the clinical outcome of NB patients. To the best of our knowledge, this study is the first to identify gene signatures related to infiltration of DCs and NK cells that better define risk stratification of NBs and strongly correlate with the expression of PD-1 and PD-L1. This invites to deepen their use for selecting patients with the greatest chance to benefit from currently available immunotherapies. In addition, these results pave the way for further studies on the mechanism underlying the establishment of T-cell-mediated antitumor immune responses in NB patients.

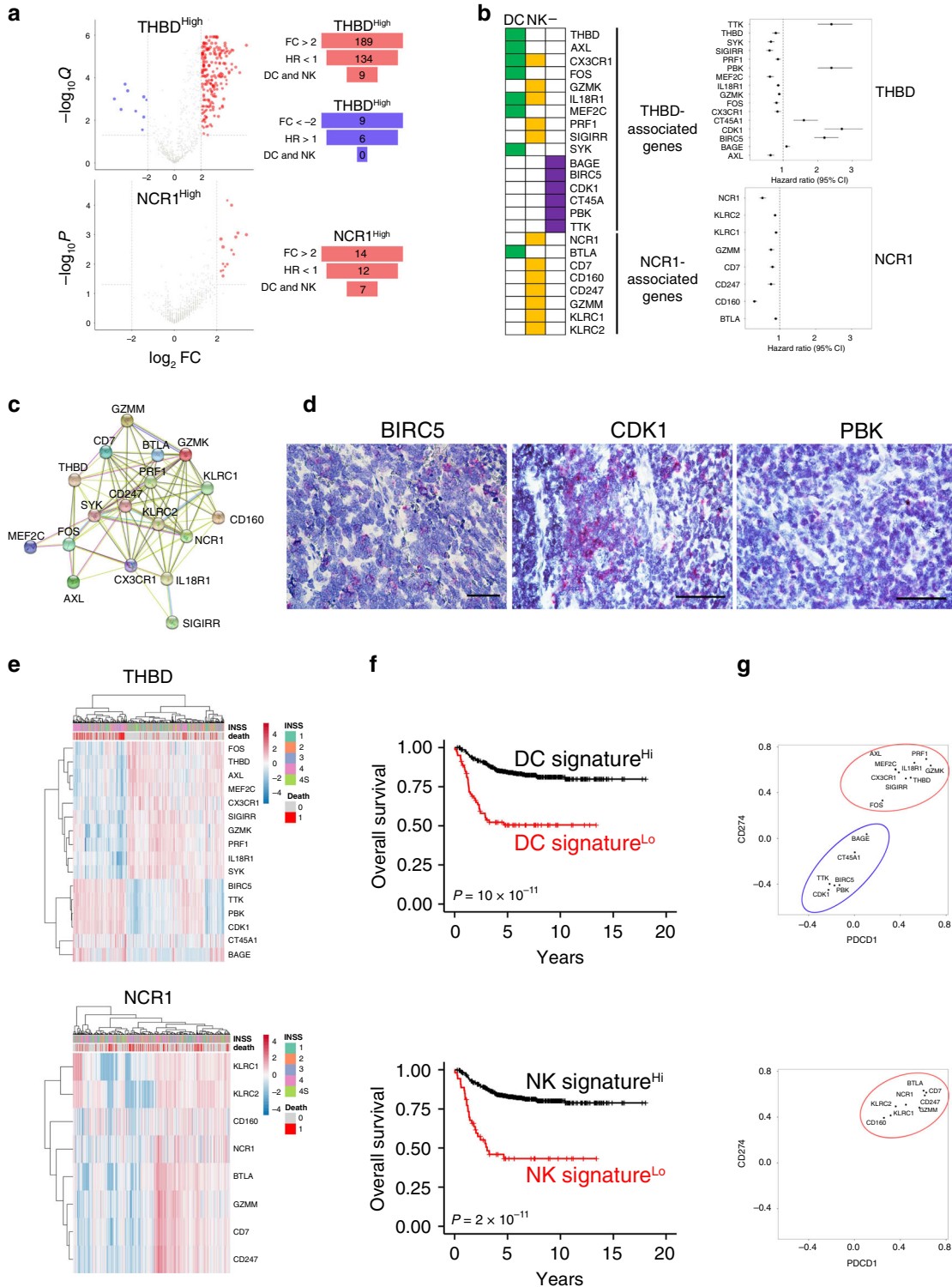

## Methods

**Patients and samples.** Tumor samples from 104 NB patients diagnosed between 2002 and 2017 at the Bambino Gesù Children's Hospital (Rome, Italy) were used. All samples were collected at diagnosis and prior to any therapy. For each patient, written informed parental consent was in accordance with the Declaration of Helsinki. The study was approved by the Ethical Committee of the Institution. Clinical pathological details of each sample are shown in Supplementary Table 1. Diagnosis and histology definition were performed according to the International Neuroblastoma Risk Group (INRG) and the International Neuroblastoma Pathology Classification (INPC)[67,68], respectively. MYCN and 1p status were evaluated following current guidelines[69]. Patients were treated according to protocols active for the different risk groups[70–73].

**Antibodies for immunostaining.** The following antibodies were used: mouse monoclonal CD141 (15C8, Leica Biosystems), and mouse monoclonal Human NKp46/*NCR1* antibodies (195314, R&D Systems) for IHC staining. Mouse monoclonal Alexa Fluor® 647 anti-human CD8a (C8/144B, BioLegend), mouse monoclonal Thrombomodulin (141C01-1009, Invitrogen), and rabbit polyclonal CD335 (NKp46, Invitrogen) antibodies were used for immunofluorescence staining. F(ab')2-goat anti-rabbit IgG (H + L, Alexa Fluor 555, Invitrogen), and F(ab')2-goat anti-mouse IgG (H + L, Alexa Fluor 488, Invitrogen) were used as cross-adsorbed secondary antibodies.

**Immunohistochemistry and acquisition.** Formaldehyde-fixed paraffin-embedded (FFPE) blocks available from 104 NB patients were cut in sections of 3-μm and

**Fig. 6 DC and NK gene signatures positively correlate with increased neuroblastoma patient survival. a** Volcano plots displaying the expression levels of immune genes according to *THBD*^high and *NCR1*^high (left panels). Genes significantly upregulated are shown in red (fold change >2, Q or P values <0.05). Genes significantly downregulated are shown in blue (fold change < - 2, Q value <0.05). A summary of the selected genes is shown in the right panels. **b** Heatmap of the genes expressed on DCs, NK cells, and other (−) cell types (left panel). Prognostic value of the indicated genes using SEQC-NB ($n =$ 498) cohort in multivariate Cox proportional hazards survival analyses (right panels). Data are expressed as a hazard ratio with 95% confidence intervals (CI); a value <1 means increased overall survival, and >1 means decreased overall survival. **c** Protein–protein interaction network for protective genes associated with either *THBD* or *NCR1* expressed by DCs and/or NK cells detected with STRING database (http://string-db.org/). The network nodes and the lines represent each protein and protein–protein associations, respectively. The color of the line indicates the type of interaction, i.e., black lines highlight the co-expression between most of the genes analyzed; the pink, blue, and green lines delineate the protein interactions validated experimentally, in silico or identified by literature, respectively. **d** Representative images selected from $n = 104$ biologically independent NB specimens. Tissue sections were probed for BIRC5, CDK1, and PBK expression using RNAscope (red dots). Nuclei were counterstained with hematoxylin (blue). Original magnification, ×40. Scale bar, 30 μm. **e** Heatmaps showing the normalized expression values of the protective and risky genes associated with either *THBD* or *NCR1* and expressed by DCs and/or NK cells in NB patients. **f** Kaplan–Meier curves for overall survival of NB patients in SEQC-NB ($n = 498$) cohort. Log-rank test with Miller and Siegmund P-value correction was used. **g** Correlation of CD274, PDCD1 expression with the indicated genes in SEQC-NB patients; red circle indicates the protective genes (FC > 2 and HR < 1), blue circle indicates the risky genes (FC < −2 and HR > 1). Correlations were assessed measuring the coefficient of determination of a robust linear regression model fit on the data (see "Statistical analysis"). Hi high, Lo low. Statistically significant P values are indicated.

baked for 60 min at 56 °C in a dehydration oven. Antigen retrieval and depar-affinization were carried out on a PT-Link (Dako) using the EnVision FLEX Target Retrieval Solution kits at high or low pH (Dako), as per the manufacturer's instruction. Following unmasking, slides were subjected to either the FLEX Per-oxidases blocking reagent (Dako) for CD141 staining (1:70 dilution) or the avidin/biotin blocking system (Thermo Fisher Scientific) for NKp46 staining (1:100 dilution) and then incubated overnight at 4 °C with primary antibodies. This step was followed by incubation with secondary antibodies coupled with either peroxidase (Dako) for CD141 or streptavidin alkaline phosphatase (Dako) for NKp46 detection. Bound peroxidase and streptavidin were detected with diaminobenzidine (DAB) solution (Dako) or Fast Red chromogen substrate (Dako), respectively. Tissue staining was counterstained with EnVision FLEX Haematoxylin (Dako). Sections of normal tonsils were used as positive controls. Isotype-matched mouse mAbs were used as negative controls. Slides were analyzed using an image analysis workstation (D-SIGHT Menarini Diagnostic). Stained slides were also scanned using the NanoZoomer S60 Digital slide scanner C13210-01 (Hamamatsu Photonics). Scanned images were viewed and captured with Hamamatsu Photonics's image viewer software (NDP.view2 Viewing software U12388-01). The density of CD141^+ and NKp46^+ cells was recorded by two blinded examiners as the number of positive cells per unit tissue surface area (mm^2). Each raw count was normalized to the tumor cellularity (ranging from 400 to 700 cells per field). The natural logarithm of the mean density of ten representative fields for each sample was used for the downstream statistical analysis. Each quantity $x$ was converted to the log scale with the formula $y = \ln(x + 1/(1 + x))$ to allow the transformation of 0 values.

**Multiplex immunofluorescence staining.** Following unmasking, FFPE slides were blocked for 60 min with 1% BSA and 5% normal goat serum and then each antibody was added consecutively as follows. First, sections were incubated with anti-thrombomodulin (CD141) mAb in 1:20 dilution overnight at 4 °C, followed by 60-min incubation with Alexa Fluor 488 goat anti-mouse IgG in 1:500 dilution. Next, slides were incubated with anti-CD335 (NKp46) Ab in 1:100 dilution overnight at 4 °C, followed by 60-min incubation with Alexa Fluor 555 goat anti-rabbit IgG in 1:500 dilution. Finally, after a 40-min incubation with mouse IgG, slides were incubated for 60 min with Alexa Fluor 647 anti-CD8 mAb in 1:600 dilution. After staining, slides were counterstained for 5 min with Hoechst (H3570, Invitrogen) and coverslipped with 60% glycerol in PBS. Confocal imaging was performed using an Olympus Fluoview FV1000 confocal microscope equipped with FV10-ASW 4.1 software, Multi Ar (458–488 and 515 nm), 2 × He/Ne (543 and 633 nm), 405-nm diode lasers and a ×60 (numerical aperture 1.42) oil objective with an electronic zoom at 2.0. Optical sections were acquired with a resolution of 1024 × 1024 pixels, a sampling speed of 12.5 μs/pixel, and 12 bits/pixel images. The density of CD8^+, CD141^+, and NKp46^+ cells, either alone or interacting with each other, was recorded by two blinded examiners as the number of positive cells per unit tissue surface area (mm^2). The mean of positive cells detected in five fields for each sample was used in the statistical analysis.

**Nanostring analysis.** Fresh-frozen NB samples available from 36 out of 104 IHC-stained NB patients were used for immune gene expression analysis. They included 11 samples of stage I, 5 stage II, 3 stage III, 14 stage IV, and 3 stage IVS. The total RNA was extracted using Total RNA Purification Plus Kit (Norgen, Biotek Corp. Thorold), and purified with RNA Cleanup and Concentration kit (Norgen, Biotek Corp. Thorold). Nanodrop 2000 (Thermo-Scientific) was used to quantify RNA concentration, whereas RNA integrity and purity were assessed with RNA Bioa-nalyzer kit (Agilent Technologies). Only samples having an RNA Integrity Number (RIN) at least seven were considered for further analysis. The expression of 730 immune-related genes and 40 housekeeping genes were assessed using the Nano-string PanCancer Immune Profiling assay (https://www.nanostring.com/products/

gene-expression-panels/gene-expression-panels-overview/hallmarks-cancer-gene-expression-panel-collection/pancancer-immune-profiling-panel), loading 100 ng of total RNA for each patient according to manufacturer's protocol (NanoString Technologies). The R/Bioconductor library NanoStringNorm[74] was used to pre-process raw Nanostring RCC files. We used the geometric mean to normalize for technical assay variation, the mean plus 2 standard deviations to estimate the background count level, and the geometric mean of the housekeeping genes to normalize for RNA content, following the manufacturer's recommendations and common practices[75]. Normalized expression values were log2-transformed. Genes were grouped into 11 immune cell type signatures (Cytotoxic CD8, Helper T cells, Dendritic cells, NK, Macrophages, Neutrophils, Memory T cells, Gamma Delta, Treg, Eosinophils, and Mast cells), derived from Bindea et al.[28]. The metagene scores for each immune cell type were calculated by taking the geometric mean of the normalized/transformed expression values of genes defined in the corresponding gene signature (log2 mean). Immune metagene scores were compared between *CD3E*^high and *CD3E*^low NB samples using the Kruskal–Wallis test. Since the immune metagenes were averaged over the signature genes, the P values were not adjusted for multiple comparisons.

**RNAscope technology.** FFPE blocks were cut in sections of 5 μm and subject to in situ detection of *BIRC5*, *CDK1*, *NCR1*, *PBK*, and *THBD* transcripts using the RNAscope assay with the corresponding probes (cat. nos. 465361, 476991, 312651, 551871, and 481241, respectively, Advanced Cell Diagnostics). The positive control probe PPIB (cat no. 313901) was included. RNAscope Intro Pack 2.5 HD Reagent Kit (red, cat no. 322372, Advanced Cell Diagnostics), was used for signal ampli-fication and RNA molecule visualization, following the manufacturer's recom-mendations. Digital images of the stained slides were acquired with NanoZoomer S60 Digital slide scanner C13210-01 (Hamamatsu Photonics) and the image analysis workstation (D-SIGHT Menarini Diagnostic).

**RNA sequencing analysis.** A publicly available (GEO accession GSE62564)[21] independent cohort of 498 NB patients transcriptionally profiled by RNA sequencing (SEQC-NB) was employed for validation of Nanostring data and as a training set for gene signatures discovery. Gene expression data were quantile-normalized using the limma R package, then corrected for batch effects by the ComBat R package before downstream analysis.

**Validation cohorts analysis.** Further bioinformatics investigation was performed by querying gene expression data from publicly available gene expression cancer datasets. Independent cohorts of 649 NB (SEQC-NB-2, GEO accession GSE45547)[32], 160 colorectal cancers (GEO accession GSE24551)[33], and 266 breast cancers (GEO accession GSE21653)[34] tissue samples, transcriptionally profiled by 44 K oligonu-cleotide microarrays, Affymetrix Human Exon 1.0 ST platform, and Affymetrix Human Genome U133 Plus 2.0 Array, respectively, were downloaded from the R2 Genomics Analysis and Visualization Platform (http://r2.amc.nl). Clinical expression analysis using genome-wide mRNA levels (Illumina mRNA-seq) and curated survival data were also downloaded from UCSC Xena hub (https://xenabrowser.net/) for TCGA SKCM and HNSC. After merging gene expression with survival data and removing samples without overall survival information, the sample sizes were 454 for SKCM and 520 for HNSC, respectively.

**Statistical analysis.** The false-discovery rate (FDR) was controlled to ≤5% using P values adjusted by the Benjamini–Hochberg method. Unsupervised complete-linkage hierarchical clustering of the log2-transformed normalized gene expression values was carried out using Pearson correlation as distance measure. Clustering results were displayed as heatmaps using log2-transformed, normalized expression

values scaled to have zero mean and unit variance. Reactome pathway analysis was performed using the ReactomePA R library[76]. Differences in the distribution of gene expression between two groups of a stratifying variable were assessed by the Kruskal–Wallis H test; comparisons involving more than two groups were assessed by Dunn's multiple comparison post hoc test. The relationship between gene pairs was assessed by fitting a robust linear regression model (RLM) (R library MASS), measuring the coefficient of determination (R2) and computing P values by a robust F-test. Overall and event-free survival curves for groups stratified by different criteria (see main text) were generated using the Kaplan–Meier method as implemented in the survival R library. Overall and event-free survival curves were compared using a log-rank test[19,77]. The survival analysis reported in Supplementary Fig. 13b and Supplementary Data 5 was conducted by splitting the SEQC-NB cohort according to a fivefold cross-validation schema repeated ten times; Kaplan–Meier survival analysis was performed on each of the held-out portions, testing for significant differences between the patient strata defined by high/low levels of the DC or NK signatures (cut points were determined by maximally selected rank statistics). Hazard ratios were estimated by Cox proportional hazards modeling (optionally with stratification by INRG) as implemented in the survival R library. The gene interaction network is based on gene–gene interaction information of the STRING database (http://string-db.org) and was generated via the query interface on their webpage. The *THBD* and *NCR1* gene signatures were refined by selecting genes predominantly expressed in DCs and NK cells using the Primary Cell Atlas database from BioGPS[30]. To assess the association between *THBD*/CD141 and *NCR1*/NKp46 expression and high/low expression of *CD3E*/CD3, a logistic regression model was trained and evaluated in a three- or fivefold CV (for IHC and SEQC-NB data) repeated ten times, assessing performance in terms of MCC. To assess the association between CD141/NKp46 expression and age at diagnosis, MYCN amplification, or INRG staging, predictive models were trained and evaluated in a threefold cross-validation schema iterated ten times. Logistic regression (for binary outcomes age at diagnosis, MYCN amplification) or multinomial logistic regression models (for multiclass outcome INRG) were used as classifiers, using R's stats::glm (with option family=binomial) and nnet::multinom functions, respectively. Predictive performance was assessed in terms of the MCC[78], which combines accuracy and precision in a single metric that has been shown to be more reliable than accuracy and F1 score[79]. MCC ranges from −1 (inverse prediction) to 1 (perfect prediction), with 0 meaning random guess. The predictive performance using the true outcome labels was compared with that of a random predictor, obtained by training the models after a random permutation of outcome labels (random labels mode). To assess the prognostic improvement achieved by the addition of the DC and NK signatures, a logistic regression model was trained with OS > 2, OS < 2 yrs as the binary outcome variable, and INRG stage, MYCN amplification status, age at diagnosis (>18 mo, <18 mo), high/low expression of *CD3E* as baseline predictors; additionally, two more models were trained by adding the DC and NK signatures. Model performance was assessed by MCC, sensitivity, and specificity. The models were trained and evaluated on the SEQC cohort in a 10 × 5-fold CV, downsampling the majority class to avoid data unbalancing. Preprocessing of raw data as well as all subsequent statistical and survival analyses were performed in the R environment for statistical computing (version 3.6.2), using libraries available on CRAN or Bioconductor repositories.

**Reporting summary**. Further information on research design is available in the Nature Research Reporting Summary linked to this article.

## Data availability

The authors declare that all data supporting the findings of this study are available within the paper and its supplementary information files. The bioinformatic investigation was performed by querying gene expression cancer datasets from GEO GSE625664 and GSE45547 for NB (DOIs: https://doi.org/10.1186/s13059-015-0694-1 and https://doi.org/10.1038/cddis.2013.84, respectively), GSE24551 for colorectal cancers (https://doi.org/10.1136/gutjnl-2011-301179), GSE21653 for breast cancers (https://doi.org/10.1007/s10549-010-0897-9), and from TCGA Research Network (https://www.cancer.gov/tcga) for SKCM and HNSC. The gene interaction network is based on gene–gene interaction information of the STRING database and was generated via the query interface on the webpage (http://string-db.org). The Primary Cell Atlas database from BioGPS (http://biogps.org) was used to refine the DC and NK-cell gene signatures. Any other relevant data and code are available from the corresponding author upon reasonable request.

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

## Acknowledgements

This work was supported by grants awarded by Fondazione AIRC per la Ricerca sul Cancro AIRC IG 2016 id. 18495 (D. Fruci), Special Project 5 × 1000 n. 9962 (F. Locatelli) and AIRC IG 2018 id. 21724 (F. Locatelli), Ministero della Salute Ricerca Finalizzata PE-2011-02351866 (D. Fruci) and RF-2016-02364388 (F. Locatelli), CAR T RCR-2019-23669115 (F. Locatelli) and Ricerca Corrente (D. Fruci and F. Locatelli), Ministero dell'Università e della Ricerca PRIN 2017WC8499_004 (to F. Locatelli), Independent Research grant AIFA 2016 (F. Locatelli). This research was also supported by two fellowships from the Fondazione Veronesi (O. Melaiu and V. Lucarini). The results shown in this study are in part based upon data generated by the TCGA Research Network: https://www.cancer.gov/tcga.

## Author contributions

D.F. coordinated the project. D.F. and O.M. conceived and designed experiments, developed methodologies, acquired, analyzed and interpreted the data, contributed to administrative, technical, or material support, and wrote the paper. O.M. performed all the experiments. M.C. preprocessed the datasets, performed statistical analyses, and generated plots. G.J. provided critical support for statistical analyses and contributed to plot generation. C.F. supervised the statistical analysis and contributed to drafting the paper. V.L. contributed to RNAscope experiments. R.B., R.D.V., A.C., and F.L. contribute to acquire and manage patients. L.A.C. contributed to acquire confocal images. V.L., L.C., V.B., and F.L. discussed the results and provided critical comments. All authors critically revised and edited the paper.

## Competing interests

The authors declare no competing interests.
