## [Peer Review File · Nature Communications]

Reviewers' comments:

Reviewer #1 (Remarks to the Author):

I have read the manuscript By Melaiu et al on gene signatures of Dendritic cells and NK cells on prognosis in paediatric populations with Neuroblastoma. They have described gene expression signatures which correlate with DC and NK cells and have used specific proteins that they have stained groups to allocate spatial location within the tumour of these cases. They conclude that DC and NK infiltration is able to predict favourable prognosis in patients and efficacy of immunotherapy.

My points requiring clarification:

1. Predicting efficacy of immunotherapy on the basis of gene expression needs a prospective trial and I do not feel the authors can be so dogmatic about the effect of their biomarkers from this study alone.
2. Fig S1A shows Heat map of high and low CD3E (n=36) but there are no dendrogram depicted (treeview) and it is not clear that all cases in the CD3E High group are indeed high for all the genes depicted. There are at least 2-3 cases that look more like the low group?
3. Was the observation of high T cell subgroups in Fig 2 stochastic? Were there more immune cell groups and T cell subgroups just because there were more immune cells in general (as depicted by the ascertainment of the cases), therefore is this just biased analysis due to numbers? Have the authors controlled for proportions of cells to determine whether there are qualitative differences in the subgroups which would be more meaningful from a functional perspective.
4. The authors mention a high correlation between CD3E and DC signature but expression levels in nanostring data for DC were much lower than NK cells and indeed for macrophages, which would argue that macrophages were probably just as important. Why focus on DCs? Indeed with the relatively small numbers of DCs representing the entire tumour tissue profiled, what were individual expression of genes in signatures and are these subject to significant dilution by cellular volume?
5. There is a very discrepant result in the Treg signature expression between SEQC data and nanostring data. The Tregs in SEQC data are the most significant difference looking at p value. Can the authors explain this discrepancy?
6. Can the authors explain the p values in Figures 2 and S2? Are these uncorrected or corrected for multiple testing? It appears they are not corrected and they should be given the large number of genes in the signatures.
7. Fig 3A staining of tissues is poor resolution and not able to be interpreted. Although it appears the difference between Hi and Lo cases may be affected by amount of stroma. The Lo case had considerably less tumour cells present compared to the Hi case? Is there a measure of cellularity to try and control for this discrepancy?
8. For any analysis of survival it needs to be clear whether the data is analysed in a univariate or multivariate model. Were other clinical factors incorporated for NB cases? For instance stage of disease or treatment provided? I note Fig S5E has THBD directly correlating with stage, are these independent or co-dependent variables? It is notable that high numbers of THBD cells correlated with good survival but also earliest INRG stage, which would be a confounder.
9. Did the Cox proportional hazards model control for all confounders in the final section. The authors mention Age, T cell infiltration, MYCN amp but what about stage in this model or

treatment?

10. I cannot make out any positive staining using NCR1 image in Fig 4. Do the Lo have any expression?

11. There are many Pearson correlations that have expression as zero that are a little worrying in their interpretation. For instance Nkp46 expression in Fig 4G has many cases at zero? The scale of CD3 and Nkp46 is also very different so I wonder whether Spearman may be better for this correlation?

12. The spatial association in Fig 5 is interesting but there are no data showing this is a consistent finding or a comparison of spatial distribution in the tumour nest compared to the tumour stroma (or away from the tumour nest)? Can the authors provide quantifiable results as well as representative images? This is important if the authors are making conclusions about interaction between cell types.

13. Furthermore, CD141 staining has considerable background with lots of green but no nuclear staining (DAPI) for co-registration. I am not sure how to interpret these images.

14. I believe the authors need to be a little more conservative about the language regarding some of their observations. In the discussion first paragraph they state "We show.. intratumoural DCs and NK cells that, similarly to T cells, markedly affected the overall survival..." This implies they have performed functional studies to show causality of effect, which has not been done in this observational study. The authors seem to occasionally misinterpret correlation for causation.

15. The authors may want to elaborate more on why using colon cancer model as a validation cohort, given that they discussed "recent studies have highlighted the existence of the NK-DC axis in adult tumours, including melanoma, breast cancer, lung cancer and head and neck squamous cell carcinoma" but not colon cancer. There are survival database for breast and lung cancers.

16. Another caution of interpretation by the authors is their link to their putative biomarker for prediction of response to immunotherapy. This was not shown here and they would need prospective data of ICI treated patients to be able to make any conclusions about this.

Reviewer #2 (Remarks to the Author):

The study of Melaiu O. et al evaluates the abundance of tumor infiltrating lymphocytes and myeloid cell types, more specifically dendritic cells (DC) and natural killer (NK) cells in localized vs metastatic tumors and their prognostic relevance. Using RNA-sequencing and microscopy-based protein detection, they demonstrate that along with high T-cell infiltration, higher abundance of NK and DC cells or NK/DC-specific expression signatures can predict a better overall survival and is associated with localized tumor stages in a cohort of 104 neuroblastoma patients. The prognostic relevance of established gene signatures was also validated in a cohort of 649 neuroblastoma and 160 colon cancer patients.

The data presented are of interest for the cancer research community, especially the pediatric oncology field, where studies evaluating the TME in larger cohorts are still scarce. The experimental setup is appropriate and the study has been carefully performed, however remains descriptive and lacks functional insights. One major concern is that DC as well as NK abundance clearly correlate with CD3e as a measure of T-cell abundance. That tumor-infiltrating T lymphocytes are associated with a better clinical outcome in neuroblastoma has been previously published by the authors (Mina M, 2015 <https://www.ncbi.nlm.nih.gov/pmc/articles/PMC4570119/>). It is thus questionable whether DC and/or NK abundance are independent prognostic markers for survival of patients with NB. As no additional functional data are presented, the overall value of the study for the broader readership

of nature communications is not clear.

Major:

1. The authors state in the abstract: "mechanisms governing the establishment of the T cell-inflamed phenotype" and later in the introduction, last paragraph: "we sought to evaluate whether these or other immune cell populations are involved in T cell-mediated antitumor immune responses to human NB". However these questions are not mechanistically addressed in the manuscript.
2. Regarding statistical analysis, how are the authors dealing with confounding factors such as stage, age etc.?
3. The gene expression analyses (published dataset Figure 1 and Nanostring Figure S1) show the same trend, but could the authors provide statistical measures to demonstrate that the same gene set is indeed statistically significant differentially expressed in CD3elo vs CD3e high tumors?
4. The authors provide survival analysis as overall survival. Event-free or progression free survival might add additional information.
5. Figure 3B: it is not clear whether this is tumor tissue as it seems to contain a large proportion of stromal cells (elongated nuclei) representing either fibroblasts or Schwann cells
6. The authors state that "high levels of THBD correlate with the NB patients survival, regardless of the T cell gene expression abundance" (page 6). This is not reflected in the survival analysis, showing that patients with high THBD do better in the presence of high CD3E as compared to those with low CD3E. This suggests that only in the presence of T-cells THBD positive cells lead to a more favorable survival.
7. Higher CD141 expression (page 7) seem to correlate with localized disease, but not necessarily with better outcome as Ms patients show varying expression, but have a good outcome. Same is true for NKp46 (page 8 and figure S6D)
8. The authors state that DCs directly interact with NK and CD8+ T-cells (page 8). This is based on single observations. Could this be quantified using image analysis methods? What would be the biological meaning of DCs interacting with NK cells in tumors?
9. The authors also claim that "In contrast, high levels of THBD/CD141 were associated with better overall survival only in samples with high levels of NCR1 (Figures 5F and 5G), thus suggesting a role of NK cells in controlling the abundance of intratumoral DCs." I don't agree with this statement, as the data presented do not allow any causal conclusions.

Minor:

1. The manuscript should be checked for English syntax and spelling, and some terms are somewhat odd e.g. page 3, bottom "good prognosis NBs" – please rephrase
2. Page 5: Table 1 not included in the manuscript
3. Fig 1A is difficult to interpret, because of bad resolution; there is no labeling of the y-axis; how were "immune genes" selected? Provide table of those genes
4. Are the p-values given in Figure 2 adjusted for multiple testing?
5. Figures 2, S2, S3, S4 are missing proper y-axis labeling
6. Fig 3a and 4a: y-axis labels are misleading
7. Figure 4b and F: stainings are hardly visible – authors could include higher magnification images in addition.
8. Figure S6B and C: correlation coefficient of NKp46 with CD8 and CD4 is rather low, so the statement on page 8 that distribution of NKp46+ cells correlated with the abundance of T-cell subsets might be questioned.
9. Page 9: "High expression of NCR1/NKp46 predicted better overall survival...": better: NKp46 was associated with...
10. Table S1: please correct: INCP -> INPC
11. MYCN gain: not clear whether these are heterogeneously amplified cases or cases with 2p gain? Do 1p aberrations include 1p loss only or other alterations on 1p?
12. Figure 6D: gene names are hardly legible and functional significance is not clear
13. Figure 6F: DCsign (CD209) is a DC associated gene – I suggest to use other terminology to describe a DC specific signature to avoid misunderstanding.

RESPONSE TO REFEREES LETTER

First of all, we would like to thank the Reviewers for their helpful criticisms and suggestions aimed at improving the quality of our manuscript. Our detailed responses to each comment of the Reviewers are provided below.

Reviewer #1 (Remarks to the Author):

I have read the manuscript By Melaiu et al on gene signatures of Dendritic cells and NK cells on prognosis in paediatric populations with Neuroblastoma. They have described gene expression signatures which correlate with DC and NK cells and have used specific proteins that they have stained groups to allocate spatial location within the tumour of these cases. They conclude that DC and NK infiltration is able to predict favourable prognosis in oatients and efficacy of immunotherapy.

My points requiring clarification:

1. Predicting efficacy of immunotherapy on the basis of gene expression needs a prospective trial and I do not feel the authors can be so dogmatic about the effect of their biomarkers from this study alone.

We thank the Reviewer for raising this point. As requested, all paragraphs addressing the possibility of predicting efficacy of immunotherapy on the basis of gene expression have been modified according to the Reviewer's recommendation (**page 2, line 13 and page 16, line 5**).

2. Fig S1A shows Heat map of high and low CD3E (n=36) but there are no dendrogram depicted (treeview) and it is not clear that all cases in the CD3E High group are indeed high for all the genes depicted. There are at least 2-3 cases that look more like the low group?

As requested by the Reviewer, new Heat maps including dendrograms have been added to the Figure S1A. The dendrogram is the result of a hierarchical clustering with complete linkage based on Euclidean distance. The two new Heat maps, related to SEQC-NB and Nanostring-NB cohorts, show a better clustering of samples (Figure S1A and S1C). Data are included in the revised manuscript (**page 5, line 6**).

3. Was the observation of high T cell subgroups in Fig 2 stochastic? Were there more immune cell groups and T cell subgroups just because there were more immune cells in general (as depicted by the ascertainment of the cases), therefore is this just biased analysis due to numbers? Have the authors controlled for proportions of cells to determine whether there are qualitative differences in the subgroups which would be more meaningful from a functional perspective.

From our experience, NBs highly infiltrated by CD3⁺ T cells (CD3 identify all T cell subtypes) are characterized by high levels of some T-cell subsets, such as cytotoxic CD8⁺ T cells (CD3⁺CD8⁺) and helper-CD4⁺ T cells (CD3⁺CD4⁺) (Mina M et al., Oncoimmunology, 2015, 4(9):e1019981, Melaiu O et al., 2017 Clin Cancer Res 2017, 23(15):4462-4472). These data were confirmed at the transcriptional level (**Figures 2**), making us confident about the robustness of our experimental procedures.

Conversely, the distribution of the other immune cells often does not reflect that of T cells, as for Treg cells, Eosinophils and Mast Cells, in which no appreciable differences are detected between *CD3E*-high and *CD3E*-low NB patients (**Figure S2**). Similarly, although both macrophages and neutrophils are abundant, they are not statistically differently distributed among patients with *CD3E*-high and *CD3E*-low, at least not as much as DC and NK cells (**Figure 2**). All these data suggest that our observations are not stochastic, but rather have a biological significance that deserves to be investigated. Moreover, gene expression levels were rigorously analyzed following the Nanostring instructions. Specifically, raw data for all genes from each sample were normalized to internal ERCC controls to eliminate technical variability of the assay. Subsequently, the counts were normalized to the geometric mean of endogenous housekeeping genes followed by \log_2 transformation. Data related to the various immune gene signatures (listed in the new supplementary Table S2) were calculated as the geometric mean of the expression levels of each constituent gene, as performed by other authors (Ayers M et al., J Clin Invest. 2017 Aug 1;127(8):2930-2940; Szekely B et al., Ann Oncol. 2018 Nov 1;29(11):2232-2239; Ghatalia P et al., J Immunother Cancer. 2019 May 28;7(1):139; Damotte D et al., J Transl Med. 2019 Nov 4;17(1):357). Further details in the Methods sections have been reported (**page 20, line 3**)

4. The authors mention a high correlation between *CD3E* and DC signature but expression levels in Nanostring data for DC were much lower than NK cells and indeed for macrophages, which would argue that macrophages were probably just as important. Why focus on DCs? Indeed, with the relatively small numbers of DCs representing the entire tumour tissue profiled, what were individual expression of genes in signatures and are these subject to significant dilution by cellular volume?

We decided to study DCs for three reasons. First, DC are the most modulated immune cell populations between *CD3E*^{high} and *CD3E*^{low}. Although, as rightly noted by the Reviewer, the expression of DCs is lower than that of NK cells and macrophages, the statistical differences between *CD3E*^{high} and *CD3E*^{low} were significantly higher in DC (1×10^{-4} and 4×10^{-30} for Nanostring and SEQC datasets, respectively) than in macrophages (2×10^{-2} and 3×10^{-4} , for Nanostring and SEQC datasets, respectively). Second, the prognostic role of macrophages has already been studied in NB (Asgharzadeh S et al. J Clin Oncol 2012; 30, 3525-32), whereas no information is currently available on the prognostic role of DCs in NB. Asgharzadeh and colleagues showed that tumor-infiltrating macrophages evaluated by the CD163 marker are more prevalent in metastatic NB. Finally, DCs are a key immune cell population critical for their ability to process and present tumor antigens to T cells (Geissmann F. Nat Immunol 2007, 8(6):558-60; Shortman K and Liu YJ. Nat Rev Immunol 2002, (3):151-61). In addition, recently their interaction with NK cells has been demonstrated to be important in adult cancers (Barry KC et al., Nat Med 2018; 24(8):1178-1191; Bottcher JP et al., Cell, 2018;172(5):1022-1037). All these data prompted us to conduct this study, never done before, on our cohort of NB patients already characterized for the density of T cells and expression of HLA class I and the immune checkpoint molecules PD-1, PD-L1 and LAG3 (Mina M. et al. Oncoimmunology 2015; Apr 2;4(9):e1019981; Melaiu O, et al. Clin Cancer Res 2017, 23, 4462-4472). Actually, as rightly noted by the Reviewer, DCs are less numerous than NK cells and macrophages, but not less powerful in inducing anti-tumor immune responses. As requested, a new table (**Table**

S2) with the expression values of all the member genes of each signature has been added.

5. There is a very discrepant result in the Treg signature expression between SEQC data and nanostring data. The Tregs in SEQC data are the most significant difference looking at p value. Can the authors explain this discrepancy?

We are aware of the discrepancy in the Treg signature between SEQC and Nanostring data, which is observed with all the T-cell subsets studied (*CD3E*, *CD4*, *CD8A*). We believe that the immune cell populations with a lower prognostic value lose their significance in the Nanostring dataset for the size of the cohort studied. Indeed, we have previously shown that tumor-infiltrating Treg cells do not have a prognostic role in NB (Mina M. et al. *Oncoimmunology* 2015; Apr 2;4(9):e1019981).

6. Can the authors explain the p values in Figures 2 and S2? Are these uncorrected or corrected for multiple testing? It appears they are not corrected and they should be given the large number of genes in the signatures.

The Reviewer is right about the fact that p values were uncorrected. The p values in Figure 2 and S2, as well as S3 and S4, 3A, 4A, 5B and 5C, are not corrected for multiple testing. This is because the metagene scores of each patient were calculated as the geometric mean expression of all the member genes of each signature and not of the individual genes separately, following common practice. As specified in the Statistical Analysis subsection, immune metagene scores were compared between *CD3E^{high}* and *CD3E^{low}* using the Kruskal-Wallis test. Since the immune metagenes were averaged over the signature genes, the p-values were not adjusted for multiple comparisons. These details have been added in the Method section of the revised manuscript (**page 20, line 7**)

7. Fig 3A staining of tissues is poor resolution and not able to be interpreted. Although it appears the difference between Hi and Lo cases may be affected by amount of stroma. The Lo case had considerably less tumor cells present compared to the Hi case? Is there a measure of cellularity to try and control for this discrepancy?

The Reviewer is right. The images were originally made at high resolution (600 dpi with Adobe Photoshop) and saved as TIFF. Unfortunately, the resolution declines when converting to PDF. To overcome this problem, each IHC and RNAscope image is now equipped with details at higher magnification. We hope the Reviewer can appreciate the new images. The previous *THBD^{low}* image has been replaced with a new one with a cellularity similar to that of *THBD^{high}*. The studied tissues are mainly stroma-poor NB, with varying levels of differentiation. Ganglioneuroblastomas, in which typical Schwann cells are evident, were excluded from the study. The density in tumor regions was recorded by two blinded examiners as the number of positive cells per unit of tissue surface (mm²). The natural logarithm of the mean density of 10 representative fields for each sample was used for statistical analysis. Each count was normalized to tumor cellularity (which ranged from 400 to 700 cells per field). This technical detail is now included in the Method section of the revised manuscript (**page 18, line 14**).

8. For any analysis of survival it needs to be clear whether the data is analysed in a univariate or multivariate model. Were other clinical factors incorporated for NB cases? For instance stage of disease or treatment provided? I note Fig S5E has THBD directly correlating with stage, are these independent or co-dependent variables? It is notable that high numbers of THBD cells correlated with good survival but also earliest INRG stage, which would be a confounder.

We thank the Reviewer for raising these points that need clarification. All survival analyses were performed with gene or protein expression data in a univariate model. No other clinical or biological factors were considered. About the question of whether *THBD* and stage are independent or co-dependent variables, we assessed statistical significance by a Kruskal-Wallis H test, which is a test for median difference. The Reviewer may be wondering instead about whether we assessed correlation/association.

We assessed differences in the variance explained by the continuous variables (CD141 and NKp46) using the discrete variables namely International Neuroblastoma Risk Group (INRG) staging system, *MYCN* amplification and age at diagnosis:

1. Homogeneity of variances across groups w.r.t. INRG stage:
 CD141 by INRG, Bartlett's $K^2=30.911$, $df=3$, $p=8.874e^{-07}$
 NKp46 by INRG, Bartlett's $K^2=54.802$, $df=3$, $p=7.568e^{-12}$
2. Homogeneity of variances across groups w.r.t. *MYCN* amplification:
 CD141 by *MYCN*, Bartlett's $K^2=10.054$, $df=1$, $p=0.00152$
 NKp46 by *MYCN*, Bartlett's $K^2=10.682$, $df=1$, $p=0.001082$
3. Homogeneity of variances across groups w.r.t. age at diagnosis:
 CD141 by age, Bartlett's $K^2=0.31042$, $df=1$, $p=0.5774$
 NKp46 by age, Bartlett's $K^2=0.33044$, $df=1$, $p=0.5654$

In summary, we tested for equality of variances across grouping variables using Bartlett's test and found that the variances were statistically different in at least two groups for INRG stage and *MYCN* amplification, while they were equal for age at diagnosis. This indicates that CD141 and NKp46 may be associated with INRG stage and *MYCN* amplification. In the manuscript, we already tested for stochastic dominance using the Dunn's test applied to CD141 and INRG, rejecting the null hypothesis for the L1-M, L2-M, L1-MS, and L2-MS pairwise comparisons. In addition, we assessed stochastic dominance by the Conover-Iman test with similar results, shown in the following tables

Conover-Iman test: comparison of CD141 by INRG groups. Reported values are the Conover-Iman t test statistics and corresponding p-values with Benjamini-Hochberg adjustment. Kruskal-Wallis chi-squared = 42.0423, $df = 3$, $p\text{-value} = 0$.			
	L1	L2	M
L2	1.0052 (0.1904)	-	-

M	7.7526 (0.0000)	4.0766 (0.0001)	-
MS	4.8232 (0.0000)	3.1662 (0.0016)	0.0373 (0.4852)

Conover-Iman test: comparison of NKp46 by INRG groups. Reported values are the Conover-Iman *t* test statistics and corresponding p-values with Benjamini-Hochberg adjustment. Kruskal-Wallis chi-squared = 43.3601, df = 3, p-value = 0.

	L1	L2	M
L2	-0.6326 (0.2642)	-	-
M	7.9552 (0.0000)	5.9531 (0.0000)	-
MS	1.5683 (0.0720)	1.7648 (0.0605)	-3.2365 (0.0017)

To further assess the association between CD141/NKp46 and INRG/*MYCN*/age, we trained a logistic regression model with CD141 and NKp46 as predictors, and INRG, *MYCN* amplification and age as response variables. To note, for INRG as response, we need to use multinomial logistic regression. The predictive models were trained and evaluated using a 3-fold cross-validation repeated 10 times, measuring the performance by the Matthews Correlation Coefficient (MCC): MCC=0 means random guess, MCC=1 perfect classification, MCC= -1 inverse classification. In this case, MCC = 0 would mean that there is no association between the predictor(s) and the response. For each response variable, we tested the individual contribution of CD141 and NKp46, as well as their combination. To compare with random prediction, each model was also trained after having randomly scrambled the response variable ("random labels" mode). The average cross-validation metrics with 95% confidence interval are reported in **Figure S9C** for each task. These new analyses are discussed in the Results and Method sections of the revised version (**page 9, line 19** and **page 22, line 6**). These data are in agreement with those previously obtained on the role of NB tissue infiltrating T lymphocytes, which were found to be predictors independent of age and *MYCN* status (Mina et al., *Oncoimmunology* 2015; Apr 2;4(9):e1019981), two prognostic factors used to stratify NB patients in risk categories (Monclair T etl., *J Clin Oncol.* 2009 Jan 10;27(2):298-303; Cohn SL et al., *J Clin Oncol.* 2009 Jan 10;27(2):289-97). In fact, CD3, as well as *MYCN* and age, are significantly associated with the stage, and consequently a strong predictor of survival. Stratifying patients at various stages is also important for making predictions about their survival. Currently, the staging of patients with NB in certain risk categories takes into account clinical parameters according to INRG staging system guidelines. Hence, as high levels of

CD3 mainly prevail in stages 1 and 2, and are correlated with better survival, the amplification of *MYCN*, as well as the age greater than 18 months (both prevalent in stages 4) correlate with a worse survival (Fig 7B, Mina et al., *Oncoimmunology* 2015; Apr 2;4(9):e1019981). This evidence indicates that the use of additional markers (such as CD141 and NKp46) associated with survival and capable of predicting the stage independently of the other clinical parameters currently available, can help the clinician to better stratify patients and predict their survival more accurately. As shown by the density plot of CD141 and NKp46, most of the deceased patients have no DCs of NK cells, confirming a strong predictive value of these markers of survival of patients with NB.

9. Did the Cox proportional hazards model control for all confounders in the final section. The authors mention Age, T cell infiltration, *MYCN* amp but what about stage in this model or treatment?

As requested by the Reviewer, the stratification by stage has been included in the Cox proportional hazards model and reported in the Results and Method sections of the revised manuscript (**page 12, line 7** and **page 21, line 26**). Regarding the treatment, as pointed out in the Method section (**page 17, line 5**), the tumor tissues were obtained at the diagnosis before any therapy to avoid any effect on the expression of the molecules to be studied.

10. I cannot make out any positive staining using *NCR1* image in Fig 4. Do the Lo have any expression?

We are aware of the low resolution of our images in the PDF file. We hope you can appreciate the new images including details at higher magnification. The sample indicated with “Lo” do not have, or have very rare, *NCR1*-expressing cells per field. The selected image includes two *NCR1*-expressing cells as control of the carried-out staining.

11. There are many Pearson correlations that have expression as zero that are a little worrying in their interpretation. For instance, NKp46 expression in Fig 4G has many cases at zero? The scale of CD3 and NKp46 is also very different so I wonder whether Spearman may be better for this correlation?

We thank the Reviewer for raising this issue. As stated in the “Statistical analysis” section (**page 21, line 21**), all pairwise relationships were assessed using a robust linear regression model, measuring the coefficient of determination (R^2). In the previous version, we had incorrectly referred to the correlation as Pearson in the figure captions and named the R as Pearson in the figures. We apologize for this mistake.

12. The spatial association in Fig 5 is interesting but there are no data showing this is a consistent finding or a comparison of spatial distribution in the tumour nest compared to the tumour stroma (or away from the tumour nest)? Can the authors provide quantifiable results as well as representative images? This is important if the authors are making conclusions about interaction between cell types.

We thank the Reviewer for this comment. Figure 5A has been revised accordingly. Specifically, we added three new supplementary figures showing i) the overview from which the details shown in Figure 5A have been extrapolated (new **Figure S7**), ii) additional examples of DC, CD8⁺ and NK cells interacting with each other (new **Figure S8**), iii) images of immunohistochemistry illustrating the spatial context in which these interactions take place (new **Figure S9**). In addition, we added a graph showing the number of DC, CD8⁺ and NK cells in close proximity to each other. All these data are discussed in the Results and Method sections of the revised manuscript (**page 9, line 7 and page 19, line 5**).

13. Furthermore, CD141 staining has considerable background with lots of green but no nuclear staining (DAPI) for co-registration. I am not sure how to interpret these images.

We thank the Reviewer for this observation. The green cells without nucleus in the upper part of the image are actually red blood cells. These cells are indicated by yellow arrows in the revised Figures 5 and in the new Figures S7 and S8.

14. I believe the authors need to be a little more conservative about the language regarding some of their observations. In the discussion first paragraph they state “We show .. intratumoural DCs and NK cells that, similarly to T cells, markedly affected the overall survival...” This implies they have performed functional studies to show causality of effect, which has not been done in this observational study. The authors seem to occasionally misinterpret correlation for causation.

We thank the Reviewer for this comment. The manuscript has been revised accordingly to these recommendations (**page 13, line 6**).

15. The authors may want to elaborate more on why using colon cancer model as a validation cohort, given that they discussed “recent studies have highlighted the existence of the NK-DC axis in adult tumours, including melanoma, breast cancer, lung cancer and head and neck squamous cell carcinoma” but not colon cancer. There are survival databases for breast and lung cancers.

We thank the Reviewer for raising the question. As requested, the prognostic value of the DC and NK signatures has been analyzed in the suggested adult cancer models. With the exception of the lung cancer, all the other adult tumor models were significant for both or only one of the two gene signatures. These data are shown in Figure S12 and discussed in the Results and Method sections of the revised manuscript (**page 11 line 19 and page 21 line 2**). Lung cancer data are not shown.

16. Another caution of interpretation by the authors is their link to their putative biomarker for prediction of response to immunotherapy. This was not shown here and they would need prospective data of ICI treated patients to be able to make any conclusions about this.

We agree with the Reviewer’s comment. The manuscript has been revised accordingly to the observations raised by the Reviewer (**page 2, line 13 and page 16, line 5**).

Reviewer #2 (Remarks to the Author):

The study of Melaiu O. et al evaluates the abundance of tumor infiltrating lymphocytes and myeloid cell types, more specifically dendritic cells (DC) and natural killer (NK) cells in localized vs metastatic tumors and their prognostic relevance. Using RNA-sequencing and microscopy-based protein detection, they demonstrate that along with high T-cell infiltration, higher abundance of NK and DC cells or NK/DC-specific expression signatures can predict a better overall survival and is associated with localized tumor stages in a cohort of 104 neuroblastoma patients. The prognostic relevance of established gene signatures was also validated in a cohort of 649 neuroblastoma and 160 colon cancer patients.

The data presented are of interest for the cancer research community, especially the pediatric oncology field, where studies evaluating the TME in larger cohorts are still scarce. The experimental setup is appropriate and the study has been carefully performed, however remains descriptive and lacks functional insights. One major concern is that DC as well as NK abundance clearly correlate with CD3e as a measure of T-cell abundance. That tumor-infiltrating T lymphocytes are associated with a better clinical outcome in neuroblastoma has been previously published by the authors (Mina M, 2015 <https://www.ncbi.nlm.nih.gov/pmc/articles/PMC4570119/>). It is thus questionable whether DC and/or NK abundance are independent prognostic markers for survival of patients with NB. As no additional functional data are presented, the overall value of the study for the broader readership of nature communications is not clear.

Major:

1. The authors state in the abstract: “mechanisms governing the establishment of the T cell-inflamed phenotype” and later in the introduction, last paragraph: “we sought to evaluate whether these or other immune cell populations are involved in T cell-mediated antitumor immune responses to human NB”. However, these questions are not mechanistically addressed in the manuscript.

We thank the Reviewer for these observations. The sentences have been changed accordingly in the revised version of the manuscript (**page 2, line 4 and page 4, line 2**).

2. Regarding statistical analysis, how are the authors dealing with confounding factors such as stage, age etc.?

We thank the Reviewer for raising this point that needs clarification. The same observation was made by the Reviewer 1 at point 8. As already reported in the answer to that point, survival analyses were performed on gene or protein expression data only, without taking into account clinical or biological variables. About the question of whether stage, age, etc, are independent or co-dependent variables, we assessed statistical significance by a Kruskal-Wallis H test, which is a test for median difference. To further assess the association between CD141/NKp46 and INRG/MYCN/age, we trained a logistic regression model with CD141 and NKp46 as predictors, and INRG, MYCN amplification and age as response variables. Data are shown in Figure S9C and discussed in the Results and Method sections (**page 9, line 19 and page 22, line 6**).

3. The gene expression analyses (published dataset Figure 1 and Nanostring Figure S1) show the same trend, but could the authors provide statistical measures to demonstrate that the same gene set is indeed statistically significant differentially expressed in CD3e-low vs CD3e-high tumors?

As requested by the Reviewer, a new table has been added (**Table S1**) which includes the list of genes studied in the Heat maps of Figures 1A and S1A. These genes are those of the Nanostring nCounter PanCancer Immune Profiling Panel (<https://www.nanostring.com/products/gene-expression-panels/gene-expression-panels-overview/hallmarks-cancer-gene-expression-panel-collection/pancancer-immune-profiling-panel>). Statistical measures (p-value and q-value) were provided for each gene, demonstrating that most of which are significantly differentially expressed in CD3E^{low} vs CD3E^{high} samples.

4. The authors provide survival analysis as overall survival. Event-free or progression free survival might add additional information.

As requested by the Reviewer, event-free survival analyses have been added in supplementary Figures **S5, S6, S9** and **S12**.

5. Figure 3B: it is not clear whether this is tumor tissue as it seems to contain a large proportion of stromal cells (elongated nuclei) representing either fibroblasts or Schwann cells

We thank the Reviewer for the observation. Actually, the previous image showed an area of stroma-rich tissue with low cellularity. Since most of the studied tissues are stroma-poor NB, with varying levels of differentiation, the image has been replaced with a more representative one enriched of tumor cells. Ganglioneuroblastomas, in which typical Schwann cells are evident, were excluded from the study.

6. The authors state that “high levels of THBD correlate with the NB patient survival, regardless of the T cell gene expression abundance” (page 6). This is not reflected in the survival analysis, showing that patients with high THBD do better in the presence of high CD3E as compared to those with low CD3E. This suggests that only in the presence of T-cells THBD positive cells lead to a more favorable survival.

We thank the Reviewer for the right observation. The text has been modified accordingly (**page 7, line 1**).

7. Higher CD141 expression (page 7) seems to correlate with localized disease, but not necessarily with better outcome as Ms patients show varying expression, but have a good outcome. Same is true for NKp46 (page 8 and figure S6D)

We thank the Reviewer for this observation. Indeed, the expression of both CD141 and NKp46 is significantly higher in the localized disease (L1 and L2), than in the metastatic disease (M). This may suggest that localization more than clinical outcome correlates with higher CD141 and NKp46 expression. However, this hypothesis is not supported by data on MS patients (MS) with a metastatic disease at good prognosis, in which the expression of CD141 and NKp46 is widely heterogeneous.

8. The authors state that DCs directly interact with NK and CD8+ T-cells (page 8). This is based on single observations. Could this be quantified using image analysis methods? What would be the biological meaning of DCs interacting with NK cells in tumors?

We thank the Reviewer for this comment. As required, a graph showing the number of DC, CD8⁺ and NK cells in close proximity to each other in highly infiltrated NB samples has been added and discussed in the Results and Methods sections of the revised manuscript (**page 9, line 7** and **page 19, line 5**). The cell-to-cell contact between DC and NK cells is critical for innate and adaptive immune responses. Although little information is available on this topic, previous studies on human and animal models have highlighted the importance of the direct contact between DC and NK cells for NK cell cytolytic activity. The authors showed that NK-cell activation by DCs was completely disrupted by trans-well separation, thus indicating that DC-mediated NK-cell activation requires intimate cell-to-cell contact (Fernandez NC et al., Nat Med, 1999; 5(4):405-411; Yu Y et al., J Immunol, 2001;166(3):1590-1600). Furthermore, direct contact between activated NK cells and DCs has been shown to improve the maturation of DCs and their ability to stimulate T-cell responses (Gerosa F et al., J Exp Med, 2002; 195 (3):327-333, 85; Vitale M et al., Blood, 2005;106:566-571). Of note, these findings have been more recently confirmed by two studies. In the first study, the authors show a key role of NK cells in the accumulation of intratumoral DCs and in the control of tumor growth (Bottcher JP et al., Cell, 2018;172(5):1022-1037). In the second study, the authors show that NK cells stably form conjugates with DCs in the tumor microenvironment and that genetic and cellular ablation of NK cells in mice results in a significant reduction of intratumoral DC (Barry KC et al., Nat Med 2018;24(8):1178-1191). Our data identify the spatial interaction of DC, T cells and NK cells in the tumor microenvironment of NB patients. This evidence, together with the identification of DC and NK gene signatures with prognostic value, suggest that DC and NK cells can be evaluated for the development of new biomarkers and therapeutic targets for NB patients. Further details have been added in the Discussion (**page 14, line 11**).

9. The authors also claim that “In contrast, high levels of THBD/CD141 were associated with better overall survival only in samples with high levels of NCR1 (Figures 5F and 5G), thus suggesting a role of NK cells in controlling the abundance of intratumoral DCs.” I don’t agree with this statement, as the data presented do not allow any causal conclusions.

We thank the Reviewer for this observation. The paragraph has been edited in the revised manuscript (**page 9, line 15**).

Minor:

1. The manuscript should be checked for English syntax and spelling, and some terms are somewhat odd e.g. page 3, bottom “good prognosis NBs” – please rephrase

We thank the Reviewer for this observation. The sentence indicated was rephrased and the entire manuscript has been carefully revised for English editing by mother tongue.

2. Page 5: Table 1 not included in the manuscript

We apologize for the mistake. Table S1 is now corrected in the revised version and corresponds to the new Table S4 (**page 17, line 8**).

3. Fig 1A is difficult to interpret, because of bad resolution; there is no labeling of the y-axis; how were “immune genes” selected? Provide table of those genes

We are sorry for the low resolution of Figures 1A and S1, due to the conversion in PDF. We hope you can appreciate them in the revised version. As requested, proper y-axis labeling has been added in both Figures. The immune genes studied are those included in the Nanostring PanCancer Immune Profiling assay. As requested by the Reviewer, a new Table (**Table S1**) including all the immune genes studied has been added in the revised version of the manuscript (**page 5, line 7**).

4. Are the p-values given in Figure 2 adjusted for multiple testing?

We thank the Reviewer for raising this point that needs to be clarified. Reviewer 1 also raised the same point. As replied to the Reviewer #1 (see answer to point 6), the p values in Figures 2, S2-S4, 3A, 4A, 5B and 5C, are not corrected for multiple testing. This is because the statistical test was performed using the average of the expression levels of all the signature genes, and not that of the individual genes. These details have been added in the Method section (**page 20, line 7**).

5. Figures 2, S2, S3, S4 are missing proper y-axis labeling

As requested, proper y-axis labeling has been added. We apologize for the lack of the Y-axis in the original version of the manuscript.

6. Fig 3a and 4a: y-axis labels are misleading

As requested, y-axis labels have been added

7. Figure 4b and F: stainings are hardly visible – authors could include higher magnification images in addition.

We are aware of the low resolution of our images in the PDF file. Our images are all made using Adobe Photoshop in the TIFF format at maximum resolution (600 dpi). Unfortunately, the resolution drops when converting to PDF. Following the advice of the Reviewer, details at higher magnification have been added to all RNAscope and IHC images in the manuscript. We hope you can appreciate the new images.

8. Figure S6B and C: correlation coefficient of NKp46 with CD8 and CD4 is rather low, so the statement on page 8 that distribution of NKp46+ cells correlated with the abundance of T-cell subsets might be questioned.

Thank you for the suggestion. The sentence has been changed (**page 8, line 13**).

9. Page 9: “High expression of NCR1/NKp46 predicted better overall survival...”: better: NKp46 was associated with...

The sentence has been changed as suggested (**page 9, line 15**).

10. Table S1: please correct: INCP -> INPC

Sorry for the mistake. INPC has been corrected in the new Table S4.

11. MYCN gain: not clear whether these are heterogeneously amplified cases or cases with 2p gain? Do 1p aberrations include 1p loss only or other alterations on 1p?

MYCN gain samples had two-three extra copies of MYCN and did not include heterogeneously amplified cases. 1p aberrations include only chromosome 1p loss. To avoid misunderstanding, “Aberration” has been replaced with “Deletion” in the new Table S4.

12. Figure 6D: gene names are hardly legible and functional significance is not clear

We thank the Reviewer for raising this point. As required, the name of genes is now written in a larger font. We used the research tool for retrieving the interacting genes database (STRING, <http://string-db.org/>) to better understand the interactions between proteins in the DC and NK signatures. The STRING database integrates several databases to provide information on possible interactions of candidate proteins. Nodes and lines have well-defined meanings. In particular, the network nodes represent proteins, while the lines represent protein-protein associations. These associations are both specific and meaningful, i.e. proteins jointly contribute to a shared function; this does not necessarily mean that they bond to each other physically. The color of the line indicates the type of interaction, i.e., the black lines highlight the co-expression between most of the genes analyzed, the pink, blue and green lines delineate the protein interactions validated experimentally, *in silico* or identified by literature, respectively. These details have been added in the figure caption of the revised manuscript (**page 26, line 18**).

13. Figure 6F: DCsign (CD209) is a DC associated gene – I suggest to use other terminology to describe a DC specific signature to avoid misunderstanding.

We thank the Reviewer for this observation. As suggested, we used the “DC signature” terminology to describe the specific DC signature.

REVIEWER COMMENTS

Reviewer #1 (Remarks to the Author):

The authors have addressed all my previous concerns.

Reviewer #2 (Remarks to the Author):

The authors have submitted a revised version of the manuscript. Most of the concerns and comments were addressed adequately and additional analysis have been included, however there remains the lack of mechanistic insight (major concern 1) and independence of NK and DC abundance as prognostic markers, as they seem to correlate strongly with CD3e+ cell abundance (major 6.).

Minor:

Page 13, lines 9-10: please rephrase as redundant: ...we found that DCs and NK cells have a predictive prognostic value...

Fig 3f: what is the difference between the two fields of view (upper and lower)?

Reviewer #3 (Remarks to the Author):

The authors identify genes characteristic of NKs and DCs that are associated with elevated CD3E expression in NB tumors and improved NB patient prognosis. Tumor specimens with elevated levels of these NK and DC-linked genes also show increased numbers of NKs and DCs and DC-T interactions by microscopy. Gene signatures of DC and NKs with possible prognostic value could be identified and were associated with expression of PD-1 and PD-L1. The authors conclude that NK and DC levels have a prognostic role and may improve risk stratification for patients. These findings provide useful insight into innate immune mechanisms in the tumor that influence patient prognosis. Some observations for consideration include:

General

1. The authors make claims about potential utility of their signatures. The claims would be stronger if they show performance characteristics in larger independent cohorts or use other methods of cross validation based on splitting the SEQC cohort into training-test subgroups. The small size of the Nanostring-NB cohort limits its ability to confirm relatively weak associations.
2. The majority of observations are correlative in nature. The authors should avoid language implying that the correlations in themselves indicate probable causation, as opposed to being consistent with prior hypotheses from pre-clinical or other studies.
3. The authors should provide a quantitative assessment of the amount of prognostic improvement that may be achieved from the addition of the NK and DC signatures to INRG stage, NMYC amplification, and/or T cell infiltration. Such assessment may include the sensitivity and specificity with respect to predicting whether patients are likely to achieve OS > 2 years.

Specific

1. Figures 1a and S1a: the heat maps as submitted don't add much information. It might be useful to annotate genes or gene clusters according to functional (eg REACTOME) category.
2. Figure 1b-e: please include quantitative criteria (eg $q < 0.05$ or the like) in the main text or figure legends for determining which pathways to include in the bar chart and scatter plots. Was the decision to present these genes driven by statistical considerations, biological plausibility, or a mix?
3. Page 7 line 11 abruptly introduces data from "deceased patients." It would be helpful for the reader to be directed in the text to more characteristics of the cohort to differentiate further between the deceased patients vs the surviving patients.

4. Page 8 line 6 and elsewhere: replace "significantly improved" with "were significantly associated with improved."
5. Page 9 lines 8,9: "CD8+ T cells were attached to DCs and more rarely with NK cells" This observation would be more compelling if it were quantitative.
6. Page 11 lines 19, 20: "The extent of this association was validated on both an independent NB dataset (SEQC-NB2)." Please clarify in the text at the beginning of this section what was the training set for the signature discovery.

Point by point response to the comments from the reviewers

Reviewer #1

The authors have addressed all my concerns.

We would like to thank the Reviewer for constructive comments and recommendations.

Reviewer #2 (Remarks to the Author):

The authors have submitted a revised version of the manuscript. Most of the concerns and comments were addressed adequately and additional analysis have been included, however there remains the lack of mechanistic insight (major concern 1) and independence of NK and DC abundance as prognostic markers, as they seem to correlate strongly with CD3e+ cell abundance (major 6).

We thanks the reviewer for constructive comments and recommendations and apologize for the misinterpretation of major concerns 1 and 6 of the previous revision.

Major concern 1

We agree with the Reviewer that elucidating the mechanistic insights governing the establishment of T cell-mediated antitumor immune responses of NB patients is of great interest. The study of the immune phenotype of tumors newly resected from NB patients represents one of the strategies for providing indications on the functional status of T cells infiltrating NB tissues with different levels of DC and NK cells. This would allow us to assess whether the presence of DC and NK cells correlates with the function of effector T cells enabling tumor growth control. Although of great interest, unfortunately we do not have the availability of fresh surgical tissues from a cohort of NB patients, nor the approval of the ethics committee for a prospective study. However, the Reviewer's considerations prompted us to evaluate the expression levels of the genes encoding for activation and cytotoxicity molecules in NB patients with different levels of CD3, DC and NK cells. Interestingly, we found a statistically significant over-representation of *IFNG*, *PRF1*, *GZMB*, *GZMH*, *GZMK*, *GZMA*, *IL7R* and *ICOS* in tissues highly infiltrated by T cells, DC and NK cells as compared to tissues with few T cells, DC and NK cells. These data were included in the revised version of the manuscript (**page 16, line 21**) and showed in the new supplementary Figures S17b and S17c. In addition, we have further edited the manuscript to avoid misinterpretations of the message provided and pointed out the need to mechanistically address this aspect in the discussion (**page 17, line 18**).

Major concern 6

We agree with the Reviewer that the abundance of DCs is strongly correlated with that of CD3 T cells, both at mRNA and protein levels (Figures 3c and 3g, respectively). The survival data shown in Figures 3e and 3i indicate that patients with high levels of DC and T cells (THBD^{high}CD3E^{high} and CD141^{high}CD3^{high}), which are the most represented at transcriptional level, had the best prognosis, whereas patients with low levels of both immune cells (THBD^{low}CD3E^{low} and CD141^{low}CD3^{low}) had poor prognosis. Of note, patients with low levels of either DC (THBD^{low}CD3E^{high} and CD141^{low}CD3^{high}) or T cells (THBD^{high}CD3E^{low} and CD141^{high}CD3^{low}) were less represented, and for this reason, they did not allow drawing solid conclusions. Similarly, the abundance of NK cells is strongly correlated with that of CD3 T cells, both at mRNA and protein levels (Figures 4c and 4g, respectively). This is reflected in the survival data (Figures 4e and 4i, respectively), in which patients with high levels of NK and T cells (NCR1^{high}CD3E^{high} and NKp46^{high}CD3^{high}) had the best prognosis, whereas patients with low levels of both immune cells (NCR1^{low}CD3E^{low} and NKp46^{low}CD3^{low}) had poor prognosis (Figures 4e and 4i, respectively). Unlike DC, high levels of NCR1 were associated with better prognosis in

patients with either high or low CD3⁺ T cells (NCR1^{high}CD3E^{high} vs NCR1^{low}CD3E^{high} pairwise log-rank *P* value = 2.5x10⁻⁸ and NCR1^{high}CD3E^{low} vs NCR1^{low}CD3E^{low} pairwise log-rank *P* value = 0.017) (Figure 4e). The sentences have been rephrased accordingly (**page 7, lines 4 and 24; page 8 line 19, page 9, line 10**).

In addition, a logistic regression model was trained and evaluated in a 10-times iterated k-fold cross-validation setting on the IHC data (k=3) and on the SEQC data (k=5) to predict CD3 (or CD3E) Hi/Lo status using either CD141 (or *THBD*) or NKp46 (or *NCR1*) as predictors. The optimal cutpoint for the Hi/Lo binary status was chosen by the maximally selected rank statistics, yielding the best dichotomic patient stratification with respect to OS. Predictive performance was assessed in terms of the Matthews Correlation Coefficient (MCC). The results were compared with the random predictor, i.e., a logistic regression model trained after having randomly permuted the target variable. A significant high association with CD3 was found for CD141 with MCC=0.50 (95% confidence interval: 0.46-0.55), as compared to the random predictor MCC=-0.02 (95% confidence interval: -0.06-0.01) (Wilcoxon rank sum test *P* < 10⁻¹⁰). Similarly, CD3 was significantly associated with NKp46 with MCC=0.42 (95% confidence interval: 0.39-0.46), that was significantly higher than the random predictor MCC=-0.06 (95% confidence interval: -0.13-0.00) (Wilcoxon rank sum test *P* < 10⁻¹⁰). These data i) suggest that the coexistence of T cells, DC and NK cells within TME represents a key feature to define the best prognosis of NBs and ii) highlight the cross-dependence between these immune players in predicting the patients overall survival. These data were included in the revised version of the manuscript (**page 7, line 26 and page 9, line 12**).

Minor:

Page 13, lines 9-10: please rephrase as redundant: ...we found that DCs and NK cells have a predictive prognostic value...

We thank the Reviewer for this comment. The sentences have been modified accordingly in the revised version of the manuscript (**page 14, line 9**).

Fig 3f: what is the difference between the two fields of view (upper and lower)?

We thank the Reviewer for raising this point that needs clarification. The two panels of Figure 3f show two representative examples of the distribution of CD141 positive cells, which can be sparsely distributed within tumor nests (upper panel), or localized within tertiary lymphoid structures (TLS) (lower panel). The caption of figure 3f has been modified accordingly (**page 26, line 12**).

Reviewer #3 (Remarks to the Author):

The authors identify genes characteristic of NKs and DCs that are associated with elevated CD3E expression in NB tumors and improved NB patient prognosis. Tumor specimens with elevated levels of these NK and DC-linked genes also show increased numbers of NKs and DCs and DC-T interactions by microscopy. Gene signatures of DC and NKs with possible prognostic value could be identified and were associated with expression of PD-1 and PD-L1. The authors conclude that NK and DC levels have a prognostic role and may improve risk stratification for patients. These findings provide useful insight into innate immune mechanisms in the tumor that influence patient prognosis.

Some observations for consideration include:

We would like to thank the reviewer for the constructive comments and recommendations that we have now carefully addressed.

General

1. The authors make claims about potential utility of their signatures. The claims would be stronger if they show performance characteristics in larger independent cohorts or use other methods of cross validation based on splitting the SEQC cohort into training-test subgroups. The small size of the Nanostring-NB cohort limits its ability to confirm relatively weak associations.

We thank the Reviewers for raising this point. Since a larger independent cohort was not available, we tested the potential utility of DC and NK gene signatures by performing a cross-validation (CV) approach on the SEQC cohort. Specifically, the data set was split by a 5-fold CV repeated 10 times. Survival analysis was performed on each of the 50 held-out portions, testing for significant differences between the patient strata defined by DC or NK signatures Hi/Lo status. These data confirming the prognostic value of these gene signatures are shown in the supplementary Figure S13b and the new Table S6 and discussed in the revised version of the manuscript (**page 12, line 13**).

2. The majority of observations are correlative in nature. The authors should avoid language implying that the correlations in themselves indicate probable causation, as opposed to being consistent with prior hypotheses from pre-clinical or other studies.

We thank the Reviewer for this comment. The manuscript has been entirely revised accordingly (**page 13, line 12; page 14, line 12; page 17, line 13**).

3. The authors should provide a quantitative assessment of the amount of prognostic improvement that may be achieved from the addition of the NK and DC signatures to INRG stage, NMYC amplification, and/or T cell infiltration. Such assessment may include the sensitivity and specificity with respect to predicting whether patients are likely to achieve OS > 2 years.

Following the Reviewer's comment, we assessed the prognostic improvement by a logistic regression model having OS>2, OS<2 as outcome variable, and INRG stage, MYCN amplification, age at diagnosis (>18 mo, <18 mo), CD3 Hi/Lo as baseline predictors. Then, we trained two more models by adding the DC and NK signatures. We assessed the model performance in terms of sensitivity, specificity, and Matthews Correlation Coefficient (MCC). The MCC is a balanced measure of accuracy and precision, ranging from -1 (inverse prediction) to 1 (perfect prediction), with 0 representing random guess. The predictive models were run in a 10x 5-fold CV using the SEQC cohort, downsampling the majority class to avoid imbalances. These new analyses are shown in the supplementary Figure S13c and the new Table S7 and discussed in the Results and Method sections of the revised manuscript (**page 12, line 18; page 23, line 1**).

Specific

1. Figures 1a and S1a: the heat maps as submitted don't add much information. It might be useful to annotate genes or gene clusters according to functional (eg **REACTOME**) category.

As requested, the genes in the heat maps were used for pathway enrichment analysis with the Reactome database and the *ReactomePA* R library (Yu G, He Q. Molecular Biosystems, 2016), testing for significance with the hypergeometric model (Boyle EI et al. Bioinformatics, 2004). All p-values were adjusted for multiple testing by the Benjamini-Hochberg correction; adjusted p-values ≤ 0.05 were deemed significant. These new

results are shown in the supplementary Figures S1 and S2 and the new Table S2, and discussed in the Results and Method sections of the revised version (**page 5, line 8; page 22, line 19**).

2. Figure 1b-e: please include quantitative criteria (eg $q < 0.05$ or the like) in the main text or figure legends for determining which pathways to include in the bar chart and scatter plots. Was the decision to present these genes driven by statistical considerations, biological plausibility, or a mix?

Figure 1b shows all biological processes derived from the gene ontology term enrichment analysis performed by DAVID Bioinformatics Resources (<https://david.ncifcrf.gov/>) resulted statistically significant after Benjamini correction ($P < 0.05$). Scatter plots showed in Figures 1c-e were selected based on both biological and statistical considerations. Gene transcripts that have been reported to be associated with an increase in immune cell infiltration, immune cell trafficking and immune functional status (Barry KC et al. Nat Med 2018, Chenf WC et al. Nat Immunol 2019, Szekely B, et al. Ann Oncol 2018, Wherry EJ et al., Nat Rev Immunol 2015) were studied in the SEQC-NB dataset. Genes shown in Figures 1c-e and S2f-h were selected based on the following criteria: i) a correlation coefficient (R) greater than or equal to at least 0.4 with statistical significance ($P < 0.05$), ii) a significantly different level of expression among patients with high and low *CD3E* expression, and iii) showing consensus in the Nanostring cohort. These details have been included in the caption figures 1b-e and S2e-h of the revised manuscript (**page 25, lines 7 and 10**).

3. Page 7 line 11 abruptly introduces data from “deceased patients.” It would be helpful for the reader to be directed in the text to more characteristics of the cohort to differentiate further between the deceased patients vs the surviving patients.

As requested, additional clinic features of the NB cohort were included (**page 7, line 14; page 9, line 1**).

4. Page 8 line 6 and elsewhere: replace “significantly improved” with “were significantly associated with improved.”

As requested by the Reviewer the sentences have been changed accordingly in the revised version of the manuscript (**page 7, lines 2, 5 and 25; page 8, line 19; page 9, line 11**).

5. Page 9 lines 8,9: “CD8⁺ T cells were attached to DCs and more rarely with NK cells” This observation would be more compelling if it were quantitative.

We thank the Reviewer for the comment. The requested information has been included in the revised version of the manuscript (**page 10, line 2**).

6. Page 11 lines 19, 20: “The extent of this association was validated on both an independent NB dataset (SEQC-NB2).” Please clarify in the text at the beginning of this section what was the training set for the signature discovery.

As requested by the Reviewer, the training set employed for signature discovery has been detailed in the revised version of the manuscript (**page 12, line 10; page 21, line 23**).

REVIEWERS' COMMENTS

Reviewer #2 (Remarks to the Author):

The authors have now adequately addressed the concerns and comments in the revised manuscript.

Reviewer #3 (Remarks to the Author):

My comments have been adequately addressed.

Response to Referees letter

Reviewer #2 (Remarks to the Author):

The authors have now adequately addressed the concerns and comments in the revised manuscript.

We would like to thank the Reviewer for constructive concerns and comments that have contributed to the improvement of the manuscript.

Reviewer #3 (Remarks to the Author):

My comments have been adequately addressed.

We would like to thank the Reviewer for constructive comments and suggestions that have helped to improve the manuscript's content.